# Synthetic extracellular matrices with tailored adhesiveness and degradability support lumen formation during angiogenic sprouting

Jifeng Liu[1], Hongyan Long [1], Dagmar Zeuschner[2], Andreas F. B. Räder[3], William J. Polacheck [4], Horst Kessler [3], Lydia Sorokin [5] & Britta Trappmann [1✉]

A major deficit in tissue engineering strategies is the lack of materials that promote angiogenesis, wherein endothelial cells from the host vasculature invade the implanted matrix to form new blood vessels. To determine the material properties that regulate angiogenesis, we have developed a microfluidic in vitro model in which chemokine-guided endothelial cell sprouting into a tunable hydrogel is followed by the formation of perfusable lumens. We show that long, perfusable tubes only develop if hydrogel adhesiveness and degradability are fine-tuned to support the initial collective invasion of endothelial cells and, at the same time, allow for matrix remodeling to permit the opening of lumens. These studies provide a better understanding of how cell-matrix interactions regulate angiogenesis and, therefore, constitute an important step towards optimal design criteria for tissue-engineered materials that require vascularization.

[1] Bioactive Materials Laboratory, Max Planck Institute for Molecular Biomedicine, Münster, Germany. [2] Electron Microscopy Unit, Max Planck Institute for Molecular Biomedicine, Münster, Germany. [3] Department of Chemistry, Technical University of Munich, Garching, Germany. [4] Joint Department of Biomedical Engineering, University of North Carolina at Chapel Hill and North Carolina State University, Chapel Hill, NC, USA. [5] Institute of Physiological Chemistry and Pathobiochemistry and Cells in Motion Interfaculty Centre (CiMIC), University of Münster, Münster, Germany. ✉email: britta. trappmann@mpi-muenster.mpg.de

Successful tissue engineering approaches rely on the availability of materials that support angiogenesis, the formation of new blood vessels from the surrounding host vasculature, in order to ensure the supply of nutrients and oxygen[1,2]. Although the formation of vascular structures has been achieved in hydrogels based on naturally occurring extracellular matrix (ECM) proteins in vitro[3–7], these matrices are not suitable for in vivo tissue repair and replacement because of their limited lifetimes, due to high enzymatic sensitivity, and insufficient mechanical strength needed to resist the high pressures and stresses present in blood vessels. Hence, synthetic materials mimicking the mechanical and biochemical properties of native ECMs while at the same time offering increased in vivo stability have been employed for implantation[8,9]. However, successful de novo vascularization of such materials has been difficult to achieve, mainly because the design criteria to develop angiogenesis-conductive materials are unknown[10].

To overcome this limitation, experimental models to study how individual ECM properties affect angiogenesis, are needed. Such model systems need to incorporate the anatomical aspects of in vivo angiogenesis, in particular, the tubular geometry of parent blood vessels that carry flow, from which vessels sprout into the surrounding matrix in response to a chemokine gradient. While microfluidic devices have been developed for this purpose[11], they commonly incorporate hydrogels based on natural ECM proteins, in particular, type I collagen and fibrin, which do not allow independent control over matrix properties[12]. In addition, such matrices do not reflect the ECM of blood vessels and are not relevant to the basement membrane that is in immediate contact with the sprouting endothelial cells[13,14]. Here, synthetic hydrogels with tunable matrix adhesiveness, stiffness, and degradability that reflect molecular and physical aspects of endothelial basement membranes could serve as ideal model substrates[15].

The complex, multi-step process of angiogenesis, which is initiated by the exit of multicellular endothelial cell strands from a parent vessel and followed by matrix remodeling to enable the formation of lumens within this strand to ultimately permit blood flow, has only been partially recapitulated in synthetic hydrogels. The formation of multicellular endothelial strands and networks has been demonstrated by us and others;[16,17] however, a fully functional endothelial tube system with perfusable lumens that connect to host vessels has not been realized. To overcome this challenge, successful model systems need to consider and reconcile the differential matrix requirements for all steps of angiogenic sprouting. One important regulator is matrix adhesiveness, which mediates interactions between endothelial cells and their underlying matrix by activating integrin signaling[18]. Integrins $\alpha_v\beta_3$ and $\alpha_5\beta_1$ have been implicated in the regulation of angiogenesis in vivo[19,20], but their involvement in the individual stages of multicellular invasion versus lumen formation has not been addressed—understanding such roles could help guide matrix design.

Here, we use a synthetic hydrogel with independently tunable biochemical and mechanical properties and integrate it into a biomimetic platform that mimics angiogenesis in vitro. We demonstrate the formation of long, lumenized, and perfusable tubes within synthetic materials that display many of the important hallmarks of in vivo blood vessels, including the correct apical-basal polarity of the endothelial cells and the basal deposition of the essential basement membrane component, laminin. Importantly, we show that such lumenized tubes only form in matrices that activate $\alpha_v\beta_3$ integrin signaling to support the initial collective migration of endothelial cells and, at the same time, can be sufficiently remodeled to facilitate the opening of lumenized and perfusable vessel structures.

## Results

**Matrix adhesiveness regulates angiogenic sprout multicellularity.** To study how matrix properties affect angiogenesis in vitro, we built on a previously established model system consisting of a tunable hydrogel embedded in a microfluidic device[16]. In this model, angiogenic sprouting from a human umbilical cord vein endothelial cell (HUVEC)-coated parent channel is initiated by a chemoattractive gradient induced by the addition of a growth factor cocktail to a parallel source channel (Fig. 1a–c). The hydrogel system is based on dextran, a protein-resistant and cell-inert polymer backbone[21]. Functionalization with methacrylates (DexMA)[22,23] or vinyl sulfone groups (DexVS)[24] enables the attachment of thiol-containing cell-adhesive peptides through Michael-type addition. Then, the material is crosslinked through matrix metalloproteinase (MMP)-sensitive dicysteine peptides derived from the cleavage site of natural type I collagen[25] (Fig. 1a), thereby, rendering it suitable for cellular remodeling, a prerequisite for functional 3D cell encapsulation.

We first used this model to investigate how the initial collective sprouting, which is required for later lumen formation, is regulated by matrix properties. Previously, we showed that changes in matrix crosslinking induce a switch in endothelial cell migratory behavior from single-cell to multicellular migration modes[16]. However, how matrix adhesiveness, an important microenvironmental cue in blood vessel formation[26], regulates endothelial cell sprouting through adhesive complexes such as the integrins is still not fully understood. To study how different densities of cell-adhesive sites regulate endothelial cell sprouting and whether this occurs through integrin signaling, we functionalized our synthetic DexMA hydrogels with varying amounts of the cell-adhesive peptide RGD, which supports $\alpha_5$ or $\alpha_v$ integrin-mediated cell attachment to selective ECM proteins[27], and is therefore relevant to in vivo interactions between endothelial cells and their underlying basement membrane[28]. We tuned RGD concentrations from 0.15 to 6 mM, the full range at which endothelial cell spreading was found to be affected on 2D surfaces (Supplementary Fig. 1). Although the speed at which endothelial cells migrated through the hydrogels was independent of RGD concentration (Supplementary Fig. 2), sprout multicellularity increased with increasing RGD concentration (Fig. 1d–h; Supplementary Figs. 3–4), suggesting that integrin activation is a regulator of multicellular endothelial cell migration. To confirm this finding, we inhibited integrin function by blocking with soluble excess RGD, whereas cells were allowed to invade into hydrogels functionalized with 6 mM matrix-bound RGD, and observed a switch to the single-cell migration mode (Supplementary Fig. 5). To demonstrate that the regulation of sprout multicellularity by matrix adhesiveness is not exclusive to HUVECs, but extends to other types of endothelial cells, we repeated the experiment with human lung microvascular endothelial cells (HMVECs), and observed collective cell migration through matrices functionalized with high concentrations of RGD, but not low (Supplementary Fig. 6).

RGD-mediated adhesion engages distinct integrins, in particular, $\alpha_v\beta_3$ and $\alpha_5\beta_1$ integrins[29] (Supplementary Table 1 and Supplementary Fig. 7), both of which have been implicated in the regulation of angiogenesis[19,20]. It was, therefore, unclear which integrin subtype was responsible for the observed multicellular migration in the RGD-functionalized hydrogels. To probe the role of $\alpha_v\beta_3$ and $\alpha_5\beta_1$ integrins individually, we modified our matrices to incorporate peptides specific for these two integrins at concentrations that affected spreading on 2D surfaces (Supplementary Fig. 8). Functionalizing our hydrogels with cyclic RGD, an $\alpha_v\beta_3$-specific peptide with higher selectivity and activity than linear RGD[30,31] (Supplementary Table 1 and Supplementary Fig. 9a), resulted in the same sprouting response as observed with

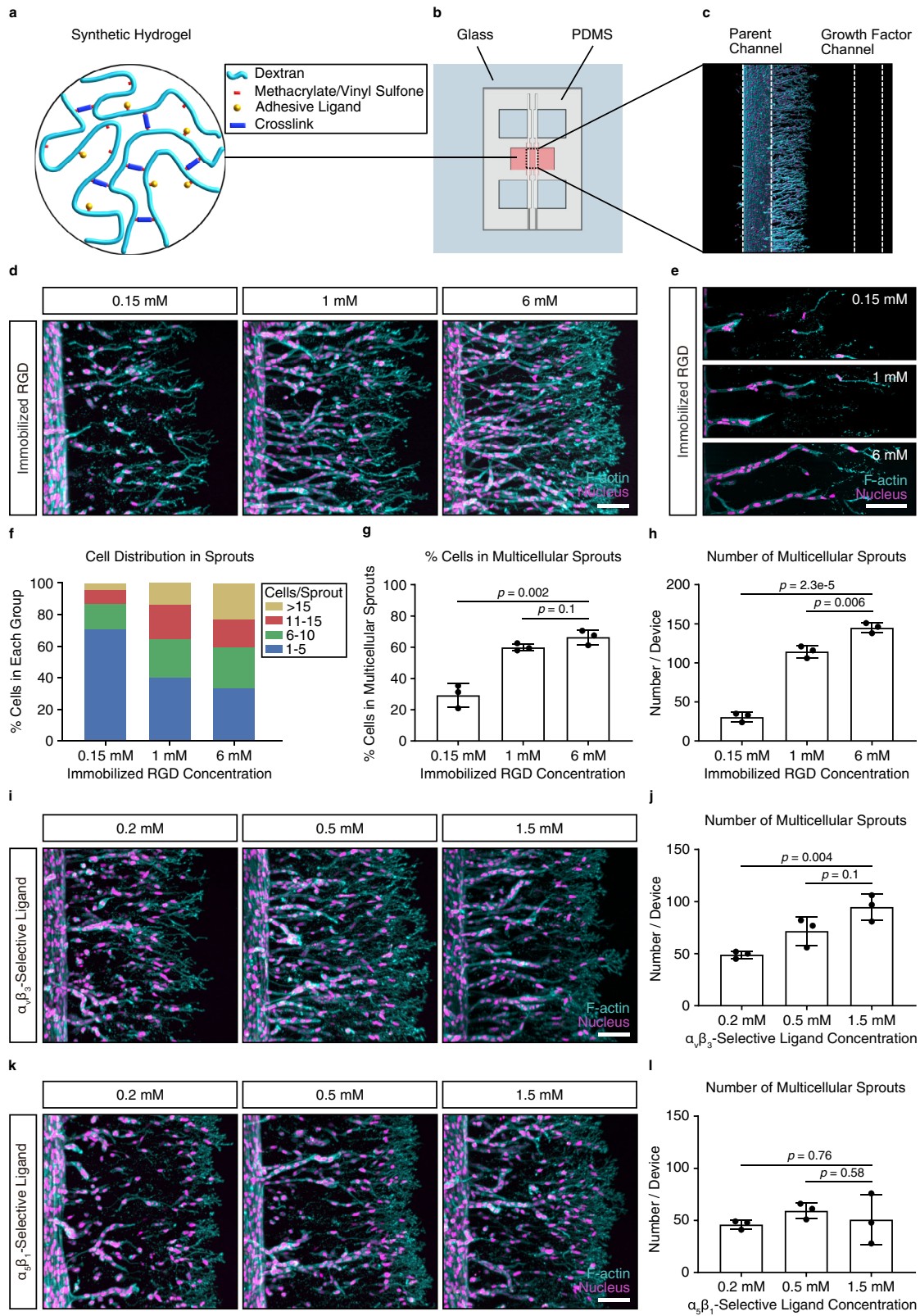

linear RGD: sprout multicellularity increased with increasing cyclic RGD concentrations (Fig. 1i, j and Supplementary Fig. 10a, b). However, when we modified matrices with an $\alpha_5\beta_1$-specific peptidomimetic ligand[32] (Supplementary Table 1 and Supplementary Fig. 9b), the migration mode was independent of ligand concentration and single-cell migration predominated at all concentrations (Fig. 1k, l and Supplementary Fig. 10c, d). To further confirm that

$\alpha_v\beta_3$, but not $\alpha_5\beta_1$ integrin activation is the main regulator of multicellularity, we allowed HUVECs to invade into matrices functionalized with 6 mM RGD (to engage both integrin types), and inhibited each integrin type individually using soluble cyclic RGD and a soluble $\alpha_5\beta_1$-specific peptidomimetic ligand. Indeed, inhibiting $\alpha_v\beta_3$ integrin engagement disrupted collective cell migration, whereas the inhibition of $\alpha_5\beta_1$ integrins did not affect

**Fig. 1 Matrix adhesiveness regulates multicellularity of angiogenic sprouting. a** Schematic of 3D dextran-based hydrogels with tunable type and concentration of adhesive ligands and crosslinkers. **b** In vitro angiogenesis model based on a microfluidic device. **c** The device consists of two parallel channels embedded within a 3D hydrogel. One channel is seeded with endothelial cells, mimicking the parent vessel. The second channel serves as a chemoattractant source, leading to the setup of a chemoattractive gradient toward the parent channel, which induces angiogenic sprouting. **d** HUVECs invading methacrylated dextran (DexMA) hydrogels functionalized with varying concentrations of the adhesive peptide CGRGDS for 3.5 days. **e** Morphology of angiogenic sprouts within the hydrogel. **f** Quantification of cell distribution across sprouts as a function of immobilized CGRGDS concentration. Sprouts were divided into four groups according to cell number per sprout. **g** Quantification of sprout multicellularity (% cells in multicellular sprouts possessing six or more nuclei, relative to the total number of cells) ($n = 3$ independent experiments). **h** Quantification of the number of multicellular sprouts (sprouts per device possessing six or more nuclei) ($n = 3$ independent experiments). **i** HUVECs invading DexMA hydrogels functionalized with varying concentrations of the integrin $\alpha_v\beta_3$-selective ligand c[RGDfK(C)]. **j** Quantification of number of multicellular sprouts (sprouts per device possessing six or more nuclei) ($n = 3$ independent experiments). **k** HUVECs invading into DexMA hydrogels functionalized with varying concentrations of an integrin $\alpha_5\beta_1$-selective ligand. **l** Quantification of number of multicellular sprouts (sprouts per device possessing six or more nuclei) ($n = 3$ independent experiments). In all experiments, coupled ligand concentration was kept constant at 6 mM by adjusting with non-adhesive ligand CGRGES. Concentration of native collagen-derived degradability (NCD) crosslinker was kept constant at 30.5 mM to ensure comparable stiffness. Composite fluorescence images of 3D projections showing F-actin (cyan) and nuclei (magenta) (scale bar, 100 μm). All data are presented as mean ±s.d., $p < 0.05$ is considered to be statistically significant (two-tailed unpaired Student's $t$ test). Source data are provided as a source data file.

multicellularity (Supplementary Fig. 11). Together, these experiments uncover $\alpha_v\beta_3$ integrins as important regulators of multicellular tissue invasion during angiogenic sprouting and demonstrate that synthetic tissue-engineered constructs must engage $\alpha_v\beta_3$ integrins to achieve successful vascularization.

**Multicellular invasion is required for lumen formation**. Although multicellular sprouting is a prerequisite of blood vessel formation and, therefore, needs to be mechanistically understood to permit better materials design, it is only the first step towards a functional vasculature; next long, lumenized, and perfusable neovessels must form from the parent vessel (Fig. 2a). Although in our experiments the collective strands of endothelial cells initially did not establish continuous polarity and were not lumenized (Supplementary Fig. 12), we hypothesized that extended culture times would allow cellular matrix remodeling and tube formation. To test this hypothesis, we initiated angiogenic sprouting in hydrogels functionalized with 12 mM RGD and, subsequently, maintained a growth factor gradient for 2 weeks to give the cells time to migrate toward the growth factor source channel and to form multicellular lumens. However, after extended culture, we noticed that the channels were narrowing, indicating that the hydrogel softened with time as a result of hydrolysis in aqueous cell culture media. To overcome this limitation, we replaced the hydrolysis-prone methacrylate groups (MA), required for coupling adhesive ligands and crosslinker peptides to the polymer backbone, with hydrolytically stable vinyl sulfone groups (VS). Although DexVS hydrogels of different adhesiveness regulated collective cell migration in a similar manner to hydrogels synthesized from DexMA (Supplementary Fig. 13), the improved stability indeed enabled long culture periods and, importantly, the formation of intracellular vacuoles and lumenized tubular structures, as visualized by perfusion with fluorescent beads (Fig. 2b, d, Supplementary Fig. 14). Endothelial cells within these tubes exhibited apical–basal polarity, as demonstrated by the correct luminal localization of the apical marker podocalyxin (Fig. 2c).

Importantly, lumenized tubes only formed in matrices that supported the multicellular migration of endothelial cells. Specifically, matrices with 12 mM bound RGD, which supported collective cell invasion, also enabled tube formation, as opposed to matrices functionalized with 0.15 mM RGD (Fig. 3a–d). Similarly, when invading endothelial cells were presented with matrix-bound cyclic RGD, selective for $\alpha_v\beta_3$, but not for $\alpha_5\beta_1$ integrins, collective sprouting resulted in the formation of lumens (Fig. 3e, f and Supplementary Fig. 15). The quality of tubes formed in hydrogels functionalized with 1.5 mM $\alpha_v\beta_3$ selective

cyclic RGD versus 12 mM linear RGD was comparable (Supplementary Fig. 16), indicating that the $\alpha_v\beta_3$ selective ligand replicates the RGD results entirely.

**Matrix degradability regulates vascular lumen formation**. To our knowledge, this is the first example of the formation of a perfusable tube in a synthetic hydrogel in vitro; however, the width of these tubes was smaller than those obtained with endothelial cells in natural ECMs, such as type I collagen (Supplementary Fig. 17)[6]. A major difference between synthetic and natural ECMs is their biochemical composition: most natural ECMs comprise flexible protein fiber networks with heterogeneous fiber sizes and micrometer-sized pores, which can be deflected or remodeled by cells, thereby, facilitating the widening of lumenized tubes. By contrast, synthetic hydrogels are uniform nanoporous materials, where the space required for tube formation relies on the active cleavage of crosslinked hydrogel molecules by cell-released proteases, such as MMPs. We, therefore, speculated that this structural difference between synthetic and natural matrices resulting in different space availability could explain the limited size and width of tubes in our hydrogel matrices. To test this hypothesis, we increased the degradability of our matrices by exchanging the crosslinker peptide and rendering it more susceptible to the cell-released MMPs (Supplementary Table 2)[33], so that matrix cleavage was no longer inhibitory, resulting in greater cell migration speed (Fig. 4a). Indeed, these high degradability (HD) matrices were able to support the formation of longer and wider lumenized tubes (Fig. 4b, e, Supplementary Fig. 17), when compared with matrices crosslinked with peptides of native collagen-derived degradability (NCD). Importantly, the formed tubes reached up to 55 μm in width, which was comparable to lumen diameters obtained in hydrogels from type I collagen (Supplementary Fig. 17). Finally, to prove that higher matrix degradability was indeed responsible for the increased size of the tubes, we inhibited the activity of cell-released MMPs using the broad inhibitor Marimastat while cells were migrating through HD matrices. This resulted in reduced speed of multicellular migration (Fig. 4f) and reduced the size of formed vessels (Fig. 4g–j), which could not be rescued even by extended culture times.

**In vitro vessels display many hallmarks of in vivo blood vessels**. After 3 weeks of culture, neovessels in HD matrices were directly connected to the parent and growth factor channels and were perfusable with fluorescent beads (Fig. 5a and Supplementary Movie 1), thereby, demonstrating further physiological

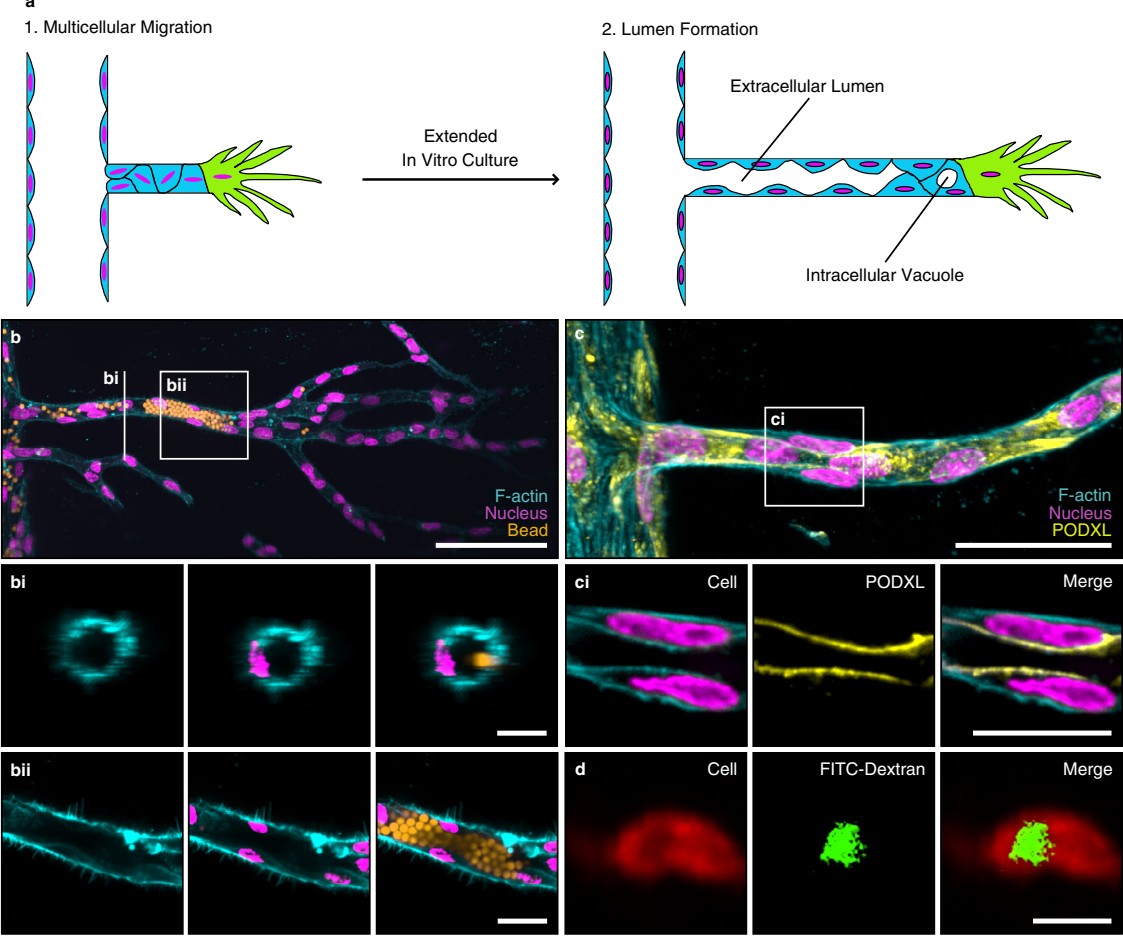

**Fig. 2 Vascular lumen formation can be achieved in synthetic hydrogels. a** Schematic of vascular lumen formation consisting of two consecutive steps: multicellular migration of HUVECs is followed by lumen formation. **b** Lumen formation in dextran vinyl sulfone (DexVS) hydrogels. Visualization of lumens by perfusion with 4 μm diameter fluorescent beads (yellow) added from the parent channel to enter the sprouts (scale bar, 100 μm), **b**i Vertical section of the 3D sprout (scale bar, 20 μm), **b**ii Horizontal section of the 3D sprout (scale bar, 20 μm). **c** Expression of the luminal marker podocalyxin (PODXL, yellow) showed the formation of intercellular lumens (scale bar, 50 μm), **c**i Horizontal section of the lumen (scale bar, 20 μm). **d** Visualization of intracellular vacuole formation by pinocytosis of FITC-dextran. FITC-dextran MW 3000–5000 Da was added in both channels for 10 days (scale bar, 20 μm). All samples contained 12 mM immobilized CGRGDS and 25.2 mM NCD crosslinker. Composite fluorescence images of 3D projections showing F-actin (cyan) and nuclei (magenta).

functionality of the newly formed vessels. To characterize the quality of flow through the neovessels, we performed time-lapse imaging of the added microparticles after applying hydrostatic pressure by adjusting the height of the fluid in the inlet and outlet ports. Although this approach only introduced relatively low flow velocities (~100 μm/s) and vessel wall shear stresses (0.4 Dyn/cm$^2$) that remained below the values measured in capillaries in vivo[34], this was due to the low driving pressures (17 Pa). However, we were able to demonstrate fully developed fluid flow, as evidenced by the parabolic microbead velocity profile (Supplementary Fig. 18). Importantly, the formed tubes showed many of the key features of in vivo blood vessels, such as the correct apical-basal polarity, as demonstrated by the basal deposition of the basement membrane-specific protein, laminin, and the localization of the apical marker podocalyxin (Fig. 5b). Analyses for endothelial-specific laminin isoforms revealed a basal deposition of both laminin 411 and 511[35,36], the latter of which is the only laminin isoform to carry an exposed RGD binding site, which is recognized by $\alpha_5\beta_1$ and $\alpha_v$-series integrins[37,38]. Electron microscopy confirmed the formation of lumenized tubes, in which individual endothelial cells were interconnected by electron-dense junctions and were basally bordered by a basement membrane-

like structure (Fig. 5c). Together, these studies emphasize the important role of ECM adhesiveness and degradability in the formation of vascular tubes: only if endothelial cells are able to sprout collectively through matrices of sufficient adhesiveness and, at the same time, of sufficient degradability, can large perfusable tubes form.

## Discussion

De novo vascularization of synthetic hydrogels through angiogenic sprouting from the host vasculature would enable the generation of tissue-engineered constructs for organ repair and replacement. Yet, the criteria for the design of such materials are still unknown, mainly owing to the lack of suitable in vitro models that mimic the structural features of native blood vessels and, at the same time, allow for the screening of individual matrix properties. Many reports using natural matrices have achieved the formation of vascular structures through angiogenic sprouting in vitro[3–7], but due to the complex composition of natural ECMs, it is not clear how individual matrix properties affect tube formation. Although recent studies have suggested that matrix density[39,40], ligand density[12], and matrix stiffness[41,42] are important regulators of angiogenesis in fibrin or collagen

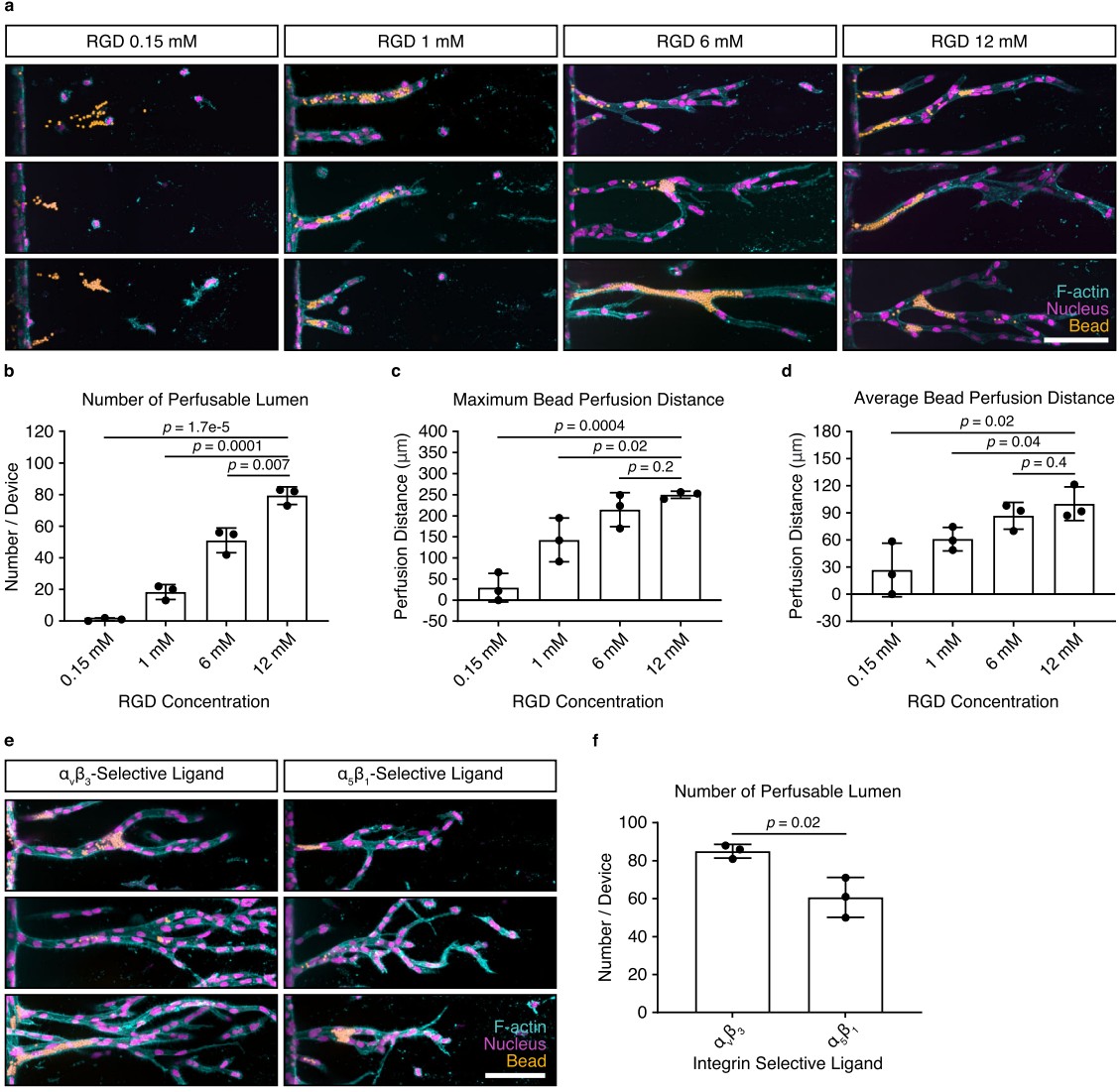

**Fig. 3 Multicellular invasion is required for vascular lumen formation. a** Lumen formation after 14 days of chemokine-guided HUVEC migration through hydrogels with different immobilized RGD concentrations. Coupled ligand concentration was kept constant at 12 mM by adjusting with non-adhesive ligand CGRGES. Concentration of NCD crosslinker was kept constant at 25.2 mM to ensure comparable stiffness. Fluorescent beads were added from the parent channel and allowed to enter the lumen by gravity. **b** Quantification of the number of lumens formed throughout the entire device, as visualized by bead perfusion ($n = 3$ independent experiments). **c** Quantification of maximum bead perfusion distance, relative to lumen opening position at parent channel ($n = 3$ independent experiments). **d** Quantification of average bead perfusion distance, relative to lumen opening position at parent channel ($n = 3$ independent experiments). **e** Lumen formation in hydrogels functionalized with 1.5 mM integrin $\alpha_v\beta_3$-selective ligand c[RGDfK(C)] and 1.5 mM integrin $\alpha_5\beta_1$-selective ligand, as visualized by the perfusion with fluorescent beads. Total ligand concentration was adjusted to 12 mM using the non-adhesive ligand CGRGES, all samples were crosslinked with 25.2 mM NCD peptide. **f** Quantification of the total number of lumens throughout the whole device ($n = 3$ independent experiments). Composite fluorescence images of 3D projections showing F-actin (cyan), nuclei (magenta), and 4 µm diameter fluorescent beads (yellow) (scale bar, 100 µm). All data are presented as mean ±s.d., $p < 0.05$ is considered to be statistically significant (two-tailed unpaired Student's $t$ test). Source data are provided as a source data file.

hydrogels, these matrix properties are intrinsically coupled in natural ECMs, making the relative contribution of each parameter difficult to determine. Using a synthetic hydrogel system with independently controllable properties, we found that the adhesiveness of matrices needs to be fine-tuned to enable the collective migration of endothelial cells and, simultaneously, matrices need to be sufficiently degradable to allow the opening of a wide, perfused lumen.

Collective endothelial cell migration is a known prerequisite of blood vessel formation[43] and we previously reported that matrix degradability switches endothelial cell migration between single-cell and multicellular modes[16]. Matrix adhesiveness is another important regulator of cell function transduced through changes

in integrin signaling[26]. Importantly, integrin signaling has been demonstrated as a key regulator of angiogenesis, and $\alpha_v\beta_3$ and $\alpha_5\beta_1$ integrins, in particular, have been implicated in successful vascularization[19,20]. However, how these integrins regulate the collective migration of endothelial cells is not known. In our studies, we find that matrices need to be functionalized with adhesive ligands that enable $\alpha_v\beta_3$ integrin engagement in order to support the formation of long, multicellular strands. At first sight, our data could appear contradictory with the many studies reporting the important role of $\alpha_5\beta_1$ integrins in the formation of vascular tubes in vivo as well as in vitro[20,44]. However, it should be noted that these reports have only examined the final stage of tube formation. In our study, we demonstrate that $\alpha_v\beta_3$ integrin is

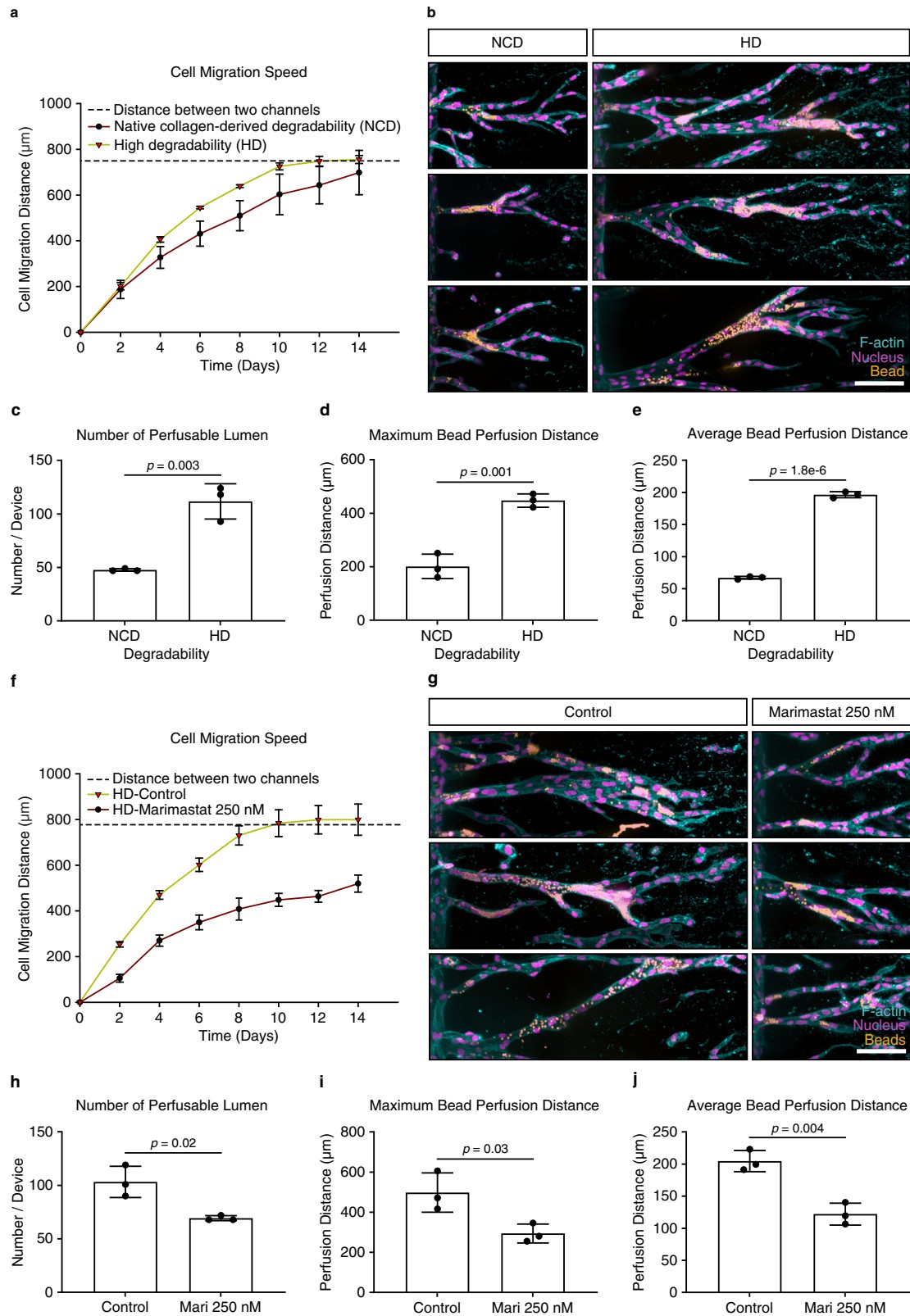

required and sufficient to drive collective endothelial cell migration in the initial step of vessel formation; but at the later tube formation step, cells have been able to deposit endogenous matrix (Fig. 5a), which also permits engagement of many other integrins, including $\alpha_5\beta_1$.

Furthermore, our results demonstrate that collective endothelial cell migration is required but not sufficient to achieve long vascularized structures in synthetic matrices. If matrix degradability is not high enough to support the remodeling by endothelial cells, which is required to make space for the formation of wide tubes, only small lumenized structures are obtained. This observation is in accordance with previous reports showing the role of matrix remodeling in the formation of vascular tubes[45]. However, it should be noted that the level of degradability

**Fig. 4 Matrix degradability regulates vascular lumen formation. a** Quantification of cell migration speed as a function of hydrogel degradability. Analyzed samples contained 12 mM immobilized CGRGDS and 25.2 mM native collagen-derived degradability (NCD) or high degradability (HD) crosslinker, respectively ($n = 3$ independent experiments). **b** Morphology of lumenized sprouts formed inside hydrogels of different degradability. Shown are three representative images each, fixed after 14 days of culture. **c** Quantification of number of perfusable lumen formed inside hydrogels of different degradability ($n = 3$ independent experiments). **d** Quantification of maximum bead perfusion distance inside hydrogels of different degradability, relative to the lumen opening position at parent channel ($n = 3$ independent experiments). **e** Quantification of average bead perfusion distance inside hydrogels of different degradability, relative to the lumen opening position at parent channel ($n = 3$ independent experiments). **f** Quantification of cell migration speed in the presence of the broad-spectrum MMP inhibitor Marimastat (250 nM) compared with non-treated controls. Analyzed samples contained 12 mM immobilized RGD and 25.2 mM HD crosslinker ($n = 3$ independent experiments). **g** Morphology of lumenized sprouts formed inside HD hydrogels in the presence or absence of 250 nM Marimastat. **h** Quantification of number of perfusable lumens inside hydrogels treated with 250 nM Marimastat relative to non-treated controls ($n = 3$ independent experiments). **i** Quantification of maximum bead perfusion distance inside hydrogels treated with 250 nM Marimastat relative to non-treated controls ($n = 3$ independent experiments). **j** Quantification of average bead perfusion distance inside hydrogels treated with 250 nM Marimastat relative to non-treated controls ($n = 3$ independent experiments). Composite fluorescence images of 3D projections showing F-actin (cyan), nuclei (magenta), and 4 μm diameter fluorescent beads (yellow) (scale bar, 100 μm). All data are presented as mean ±s.d., $p < 0.05$ is considered to be statistically significant (two-tailed unpaired Student's $t$ test). Source data are provided as a source data file.

requires fine-tuning. Although high matrix degradability is advantageous for large tube formation, we previously reported on the importance of lower degradability matrices for the collective sprouting of endothelial cells, as very degradable matrices result in single-cell migration[16]. Taken together, these studies demonstrate that balanced degradability levels are essential to fulfill the matrix requirements for both collective migration and subsequent tube formation.

Some examples of in vitro lumen formation in synthetic hydrogels have been reported[46–48], most of which were based on self-sorting of hydrogel-embedded endothelial cells and resulted in small lumens or fused vacuoles within individual cells which superficially appeared as tubes. Therefore, the resulting structures were usually not functional in transporting oxygen and nutrients because they lacked the connection to a parent vessel required for perfusion. To overcome this issue, our approach was to employ tunable hydrogels as a means of exploiting angiogenic processes for their vascularization. The results obtained from our model system will be directly applicable to the generation of tissue-engineered constructs, because they provide matrix design criteria for the incorporation of a functional, i.e., perfusable, vasculature in a cell-free material system, which would not rely on patient-derived material of limited availability.

Besides potential applications in matrix design for tissue engineering and regeneration, the presented model could have broad utility in studying the causes and progression of diseases with vascular defects in well-controlled microenvironments. Although some in vitro angiogenesis models have been able to recapitulate structural features of native parent- and neovessels[3–7], these models are mostly based on natural ECMs; owing to their fibrous nature, such ECMs are structurally and mechanically complex, in turn impacting cell function[49]. Furthermore, natural ECMs contain highly glycosylated adhesive molecules that can interact with bioactive molecules and proteins present in the cell culture medium, thereby, affecting cell function in non-definable manners and complicating the analysis of target parameters. Furthermore, a well-controlled angiogenesis model, such as the one presented here, could greatly contribute to the growing field of organoid models, whose progress is still impeded by the lack of vascularization, as this factor limits tissue size. Together, the established in vitro model should be applicable to many other biological or biomedical questions that depend on vascularization.

## Methods

**Reagents**. All reagents were purchased from Sigma Aldrich and used as received unless otherwise stated.

**Peptides and cell-adhesive ligands**. The cell-adhesive ligand CGRGDS, non-adhesive ligand CGRGES and crosslinker peptides of native collagen degradability CGPQGIAGQGCR (NCD-CR), KCGPQGIAGQCK (NCD-KK), KCGPQGIAG-QACK (NCD-KAK), and highly degradable crosslinker peptide KCDGVPMSMRGGCK (HD) were custom synthesized (provided as HCl salt) by Genscript at >95% purity. The cell-adhesive integrin $\alpha_v\beta_3$-selective ligand c[RGDfK (C)] was ordered from Peptides International (RGD-3794-PI), and the cysteine-functionalized integrin $\alpha_5\beta_1$-selective ligand was synthesized following a literature protocol[32,50].

**Antibodies**. The mouse anti-human podocalyxin antibody (10 μg/mL) was purchased from R&D Systems (MAB1658). Rabbit anti-collagen IV antibody (1:100) was obtained from Abcam (ab6586). Mouse anti-integrin $\alpha_v\beta_3$ antibody (20 μg/mL) was purchased from Merck Chemicals (MAB1976Z). Rabbit anti-laminin (1:500), rabbit anti-laminin 411 (1:500), rabbit anti-laminin 511 (1:500), and mouse anti-integrin β1 (20 μg/mL) were prepared by the laboratory of Lydia Sorokin. The secondary antibodies Alexa Fluor 555 donkey anti-mouse IgG (1:200) and Alexa Fluor 647 goat anti-rabbit IgG (1:200) were purchased from Thermo Fisher (A-31570 and A-21244, respectively).

**Synthesis of DexMA**. Methacrylated dextran (MP Biomedicals, MW 86,000 Da) was prepared according to a previously published procedure[22]. In brief, dextran (20 g) and 4-dimethylaminopyridine (2 g) were dissolved in anhydrous dimethyl sulfoxide (100 mL). Glycidyl methacrylate (24.6 mL) was added under stirring, the mixture was heated to 45°C, and the reaction allowed to proceed for 24 h. Next, the solution was precipitated into cold 2-propanol (1 L), the crude product was collected, re-solubilized in Milli-Q water, and dialyzed against Milli-Q water for 3 days. A methacrylate/dextran repeat unit ratio of 0.7 was determined by ¹H-NMR spectroscopy.

**Synthesis of DexVS**. Vinyl sulfone functionalized dextran (MP Biomedicals, MW 86,000 Da) was prepared following a modified literature protocol[51]. In brief, divinyl sulfone (4.96 mL) was added to an aqueous solution of sodium hydroxide (0.1 M, 200 mL) and dextran (4 g) under vigorous stirring at room temperature. After 5 min, the reaction was stopped by the addition of hydrochloric acid to adjust the pH of the solution to 5. The crude product was then purified by dialysis (SnakeSkin™ Dialysis Tubing, Life Technologies, 10 kDa MW cutoff) against Milli-Q water for 3 days, with two water changes daily and the final product was recovered by lyophilization. ¹H-NMR analysis in D₂O revealed a vinyl sulfone/dextran repeat unit ratio of 0.63.

**Preparation of MMP-cleavable DexMA hydrogels**. MMP-cleavable DexMA hydrogels were used for all multicellularity studies in Fig. 1. Their synthesis followed a previously published procedure[16]. A solution of DexMA (final concentration of 4.4% w/v) and variable concentrations and types of adhesive ligand (adjusted to a total concentration of 6 mM using the non-adhesive ligand CGRGES) was prepared in a solution of M199 media and HEPES (10 mM). The pH was adjusted to 8.0 using 1 M NaOH and the adhesive ligand allowed to couple to DexMA for 30 min. Then, 30.5 mM NCD-CR crosslinker was added and the gelation was initiated by readjusting the pH to 8.0. The hydrogel precursor was immediately pipetted into the central chamber of the microfluidic device, allowed to polymerize for 30 min at room temperature, and covered with phosphate-buffered saline (PBS) solution.

**Preparation of MMP-cleavable DexVS hydrogels**. MMP-cleavable DexVS hydrogels were used for all lumen formation studies in Figs. 2–5. A solution of DexVS (final concentration of 4.2% w/v) and variable concentrations and types of

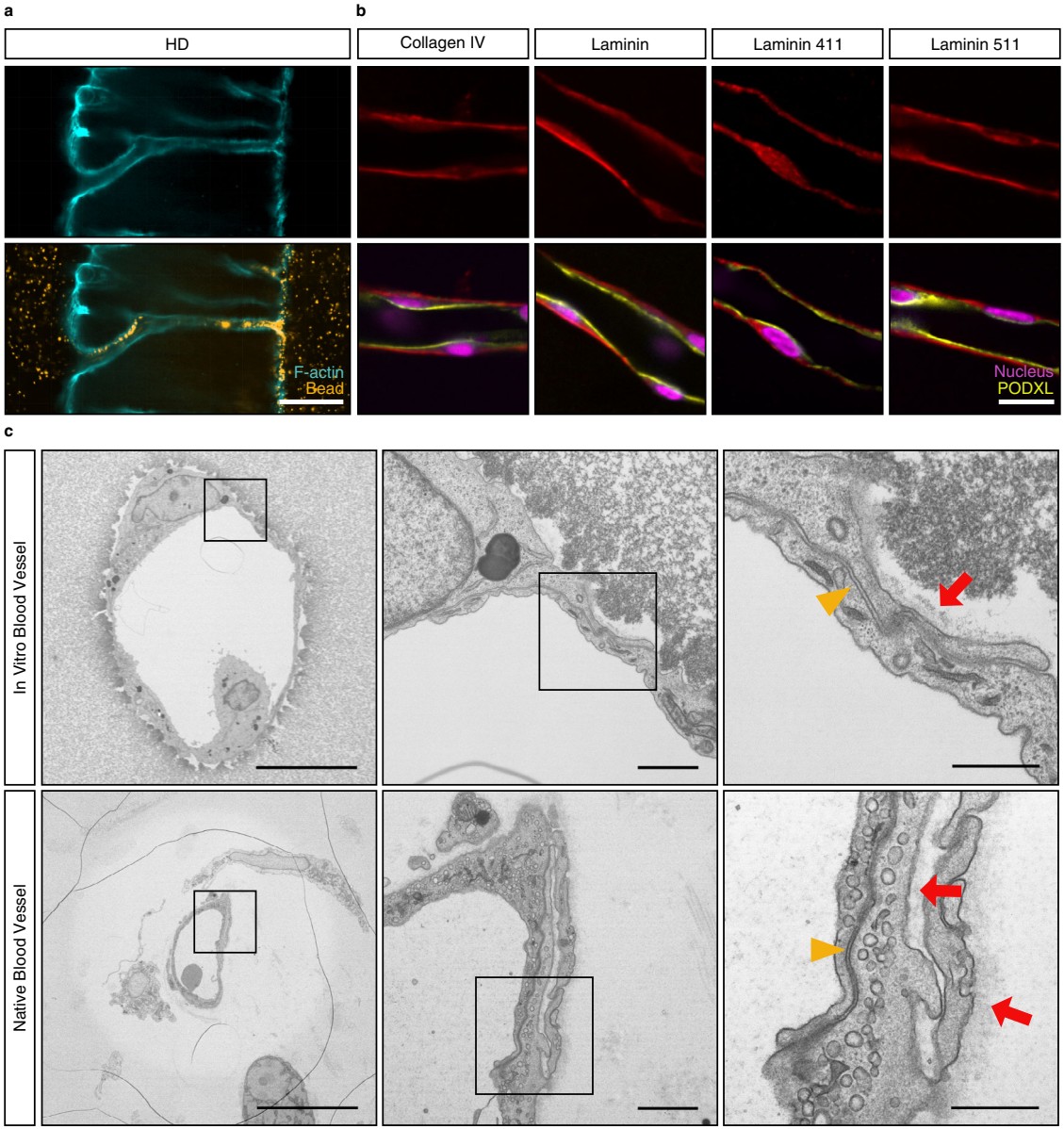

**Fig. 5 In vitro vessels display many hallmarks of in vivo blood vessels. a** Neovessels connecting parent and growth factor source channels after 3 weeks of culture in the presence of chemokine gradients. Flow through neovessels visualized by the perfusion with 1 μm diameter fluorescent beads (scale bar, 100 μm). **b** Basement membrane deposition, as visualized by immunofluorescence staining for collagen IV, pan-laminin, laminin 411, and laminin 511 (red), in combination with correct apical-basal polarity, as demonstrated by the expression of the apical marker podocalyxin (PODXL, yellow). Nuclei counterstained in magenta. (scale bar, 20 μm). **c** Comparison of in vitro and native vessels by electron microscopy reveals strikingly similar morphologies with regards to junctional integrity (yellow triangle), endogenous ECM deposition (red arrow), and cell-flattening (scale bar, 10 μm, 1 μm, 500 nm). All samples contained 12 mM immobilized CGRGDS and 25.2 mM high degradability (HD) crosslinker.

adhesive ligands (adjusted to a total concentration of 12 mM using the non-adhesive peptide CGRGES) was prepared in PBS on ice. To initiate the coupling of adhesive ligands to DexVS, the pH of the solution was adjusted to 8.0 using 1 M and 0.1 M NaOH. The solution was allowed to react for 30 min at room temperature, followed by the addition of variable concentrations of crosslinker peptides (25.2 mM NCD-KK for gels in Figs. 2, 3; 25.2 mM NCD-KAK and 25.2 mM HD for gels in Fig. 4; 25.2 mM HD for gels in Fig. 5). The solution was again cooled on ice, and the pH was readjusted to 8.0 using 0.2 M NaOH to initiate hydrogel cross-linking. The solution was immediately transferred to the central chamber of the microfluidic device and allowed to gel for 30 min at room temperature, followed by hydration in PBS.

**Mechanical testing of hydrogels**. Young's moduli of DexMA and DexVS hydrogels were characterized using a nanoindenter (Piuma, Optics 11, Netherlands). All hydrogels used in this study had a Young's modulus of ca. 1 kPa (data in Supplementary Fig. 19).

**Angiogenic device fabrication**. Microfluidic devices mimicking sprouting angiogenesis in vitro were prepared as previously described[6,16]. In short, the device housing consisted of two patterned layers of PDMS (Dow Corning, 10:1 base: curing agent), which were molded from photographically patterned silicon masters, sealed against a glass coverslip, and sterilized using UV light. To form tubular channels, two acupuncture needles (Hwato, 400 μm diameter) were coated with a 5% (wt/vol) aqueous solution of gelatin, cooled to 4 °C, and inserted into the UV-sterilized device. The hydrogel solution was added to the central chamber of the device and allowed to polymerize for 30 min at room temperature. The resulting gels were hydrated in PBS overnight at 37 °C to melt the gelatin coating, followed by needle extraction and thorough washing with PBS and EGM-2 prior to cell seeding.

**Cell culture**. HUVECs and HMVECs were purchased from Lonza (catalog number C2519A for HUVECs and CC-2527 for HMVECs) and cultured in fully supplemented EGM-2 (for HUVECs) or EGM-MV (for HMVECs) (PromoCell) containing additional 250 ng/mL amphotericin B and 10 μg/mL gentamicin (Gibco).

Cell cultures were maintained in incubators with constant humidity at 37 °C and 5% $CO_2$. In all angiogenesis assays, cells from passage 4–6 were used.

**2D cell attachment studies**. DexMA hydrogels functionalized with varying amounts and types of adhesive ligands and crosslinked with 30.5 mM NCD-CR peptide were prepared on glass coverslips. Freshly trypsinized HUVECs were suspended in EGM-2 and seeded on the hydrogel surfaces at a density of 20,000 cells/cm². Cells were allowed to adhere and spread for 24 h prior to fixation. The concentrations supporting maximum cell spreading were determined for each ligand individually. Those concentrations were then used as the highest conditions in the following sprouting assays.

For integrin inhibition experiments, HUVECs were suspended in EGM-2 containing varying concentrations and types of integrin blocking ligands and antibodies, followed by seeding (density of 25,000 cells/cm²) on DexMA hydrogels functionalized with 6 mM CGRGDS, 1.5 mM integrin $\alpha_v\beta_3$-selective ligand c [RGDfK(C)], or 1.5 mM integrin $\alpha_5\beta_1$-selective ligand (adjusted to a total concentration of 6 mM using the non-adhesive ligand CGRGES), and crosslinked with 30.5 mM NCD-CR peptide. Cells were allowed to attach and spread on the surface of the hydrogel for 2 h prior to fixation.

**Angiogenesis assays in synthetic hydrogels**. A solution of 10 million cells/mL EGM-2 (for HUVECs) or EGM-MV (for HMVECs) was added to one reservoir and cells were allowed to attach to the bottom side of the channel for 30 min, followed by seeding of the top channel side for another 30 min. Culture medium was exchanged and cells were allowed to adhere and spread on the channel walls for 1 h. Then, the medium was exchanged and reservoirs were scratched to remove any unattached cells. Finally, culture medium containing 150 ng/mL phorbol 12-myristate 13-acetate (PMA) was added to the growth factor channel, and the devices were placed on a platform rocker (BenchRocker BR2000) to initiate gravity-driven flow. 24 h later, a growth factor cocktail consisting of 75 ng/mL vascular endothelial growth factor (rhVEGF 165, R&D Systems), 150 ng/mL PMA (Sigma), and 250 nM sphingosine-1-phosphate (S1P, Cayman Chemical) in culture medium was added to the growth factor channel to induce angiogenic sprouting. Medium and cocktail were exchanged daily. Cells were fixed at a constant invasion depth (halfway between the parent and growth factor channel, reached after 3.5 days of cocktail treatment) for multicellularity experiments. Lumen formation experiments required longer culture times for cells to sufficiently remodel the matrix, and samples were fixed after 14 days.

For MMP inhibition experiments, Marimastat (Tocris Bioscience) was added to the growth factor cocktail at 250 nM. Blocking of integrins with soluble adhesive ligands during angiogenic sprouting was achieved by the addition of the respective ligands to the growth factor cocktail solution.

**Angiogenesis assays in type I collagen hydrogels**. The PDMS and glass surfaces of the microfluidic chamber were functionalized with an aqueous 0.1% (w/v) poly-L-lysine and subsequently 1% (w/v) glutaraldehyde solution. To form tubular channels, two acupuncture needles (Hwato) of 400 µm diameter were coated with a 0.4% aqueous solution of bovine serum albumin (BSA), sterilized using UV light, and inserted into the device. A 2.5 mg/mL collagen type I (rat tail, Corning) solution was cast inside the device and allowed to polymerize for 30 min at 37°C. The resulting gels were hydrated in PBS overnight and washed thoroughly with PBS following needle extraction. A solution of 1 million cells/mL medium was added to a reservoir and cells were allowed to attach to the bottom side of the channel for 10 min, followed by seeding of the top channel side for another 10 min. After washing with medium, the devices were placed on a platform rocker to initiate gravity-driven flow. On the second day, a growth factor cocktail consisting of 75 ng/mL vascular endothelial growth factor (VEGF) (R&D Systems), 10 ng/mL PMA (Sigma), and 500 nM S1P (Cayman Chemical) in a culture medium was introduced to the second channel to induce angiogenic sprouting. Medium and cocktail were exchanged daily until cells were fixed in 4% paraformaldehyde after 9 days of culture.

**Fluorescent staining and microscopy**. Cells on 2D hydrogel surfaces were fixed with 4% paraformaldehyde (Thermo Fisher) for 15 min at room temperature, followed by permeabilization with 0.5% Triton-X100 in PBS for 10 min. Samples were incubated with Alexa Fluor 488 Phalloidin (1:200, Thermo Fisher) and Hoechst 33342 (1:200, Life Technologies) in PBS for 1 h at room temperature. The number of attached cells was automatically counted in ImageJ (v2.0.0-rc-68/1.52e).

Cells in devices were fixed with 4% paraformaldehyde (Thermo Fisher) for 1 h at room temperature, followed by permeabilization and blocking with 0.5% Triton X100 in 3% BSA for 1 h at room temperature. Samples were incubated with primary antibodies (diluted in 3% BSA) overnight at 4°C, washed thoroughly with PBS, and incubated with Alexa Fluor (555 and 647)-conjugated secondary antibodies overnight at 4°C. Samples were washed with PBS for 2 days with multiple buffer exchanges. Next, cells were stained with Hoechst 33342 (1:200, Life Technologies) and Alexa Fluor 488 Phalloidin (1:200, Thermo Fisher) overnight at 4°C. If antibody staining was not required, samples were directly stained with phalloidin and Hoechst 33342 after permeabilization. All incubation steps were performed on a platform rocker to facilitate the transport of antibodies and dyes into the gel.

Fixed and stained devices were imaged at 10× (sprout overview) or 40× (individual sprouts) using a spinning disc confocal microscope (Dragonfly by Andor with built-in software Fusion, v2.0.0.13). Images are presented as maximum intensity projections. For quantification of multicellular sprouting, surfaces were constructed in IMARIS (vx64, 9.5.1). Sprout multicellularity was analyzed by manually counting the number of nuclei per sprout structure. Sprouts with at least six nuclei were defined as multicellular. The number of cells in multicellular sprouts was presented relative to the total number of cells. Lumen formation was analyzed by perfusion with fluorescent beads. Lumens longer than 20 µm and bead perfusion distances were counted manually in IMARIS. Bar graphs were generated with Sigmaplot (v14.0) or Graphpad Prism (v9.0.0).

Brightfield imaging was performed on a Leica DMi1 inverted microscope (with built-in software Leica application suite, v3.4.0).

**Endothelial cell vacuole visualization**. Staining of vacuoles inside HUVECs followed a literature protocol[52]. In brief, lentivirally infected HUVECs expressing a red fluorescent protein (LifeAct-mRFPruby, cells donated by the lab of Christopher Chen at Boston University, USA) were seeded in microfluidic devices and sprouting was induced for 10 days. Throughout the culture period, media in the HUVEC channel, as well as the growth factor cocktail, were supplemented with an additional 0.1 mg/mL fluorescein isothiocyanate-dextran 3000–5000. Finally, samples were fixed with 4% paraformaldehyde for 1 h at room temperature and washed thoroughly for 2 days on an orbital shaker including several PBS exchanges prior to image acquisition by confocal microscopy.

**Generation of neovessels connecting parent and growth factor source channel**. To generate neovessels that are able to support flow through the hydrogel, curved acupuncture needles were inserted into the device housing to shorten the distance to 330 µm between the parent and growth factor source channel. Hydrogel casting and HUVEC seeding were performed as described before. Sprouting was induced for 21 days allowing the formation of long lumens spanning the two channels. The flow was visualized through the addition of 1.0 µm diameter red fluorescent beads (FluoSpheres Sulfate Microspheres, Invitrogen) to the growth factor source channel prior to the acquisition of a video using a confocal microscope.

**Perfusion experiment and determination of vessel wall shear stress**. To determine wall shear stress within neovessels, 1 µm fluorescent microparticles were introduced in the upstream vessel and hydrostatic pressure was applied by adjusting the height of the fluid in the inlet and outlet ports. Streaks of flowing microparticles were captured by acquiring images with 200 ms exposure time, and the length of the streak and distance from the vessel wall were measured in ImageJ (v2.0.0-rc-68/1.52e). From the flow profile measured at the midpoint of a neovessel and the local vessel diameter, the volumetric flow rate and average fluid velocities were computed. After verifying the characteristic parabolic flow profile of Poiseuille flow, the wall shear stress was computed assuming the viscosity of water at 37°C. For an applied pressure of 17 Pa in the inlet channel, a wall shear stress of 0.428 Dyn/cm² was calculated.

**Mouse cremaster tissue**. Mouse cremaster tissue was obtained from 10-week-old C57/BL6 wild-type (Charles River Laboratories), female mice. Mice were maintained in individually vented cages (temperature between 21.5 and 22.5°C, 50–60% humidity) under a 14 h light/10 h dark cycle with free access to food and water. Animal husbandry was performed according to the German Animal Welfare guidelines and approved by the Landesamt für Natur, Umwelt und Verbraucherschutz Nordrhein-Westfalen (State Agency for Nature, Environment and Consumer Protection of North Rhine-Westphalia, Az 84-02.05.20.13.025).

**Electron microscopy**. HUVECs cultured in microfluidic chambers were fixed with 2% glutaraldehyde and 2% paraformaldehyde in 0.1 M cacodylate buffer, pH 7.4 for 3 h at room temperature. Subsequently, the cell-containing hydrogel was removed from the device and trimmed into smaller parts with a razor blade before further processing. The sample was post-fixed in 1% osmium tetroxide, containing 1.5% potassium cyanoferrate, and cut further into smaller cubes while keeping the original orientation. The specimen was en-bloc stained overnight in 0.5% uranyl acetate in 70% ethanol and stepwise dehydrated in ethanol to 100%. The last drying steps were performed in propylene oxide, and the sample was infiltrated with mixtures from propylene oxide and epon and finally pure epon (partially under vacuum for better infiltration). The samples were orientated in flat embedding molds and hardened at 60°C. 60 nm ultrathin sections were cut on an ultramicrotome (UC6, Leica) and placed on one slot formvar coated copper grids. The sections were counterstained with lead and analyzed at 80 kV with a transmission electron microscope (Tecnai-12-biotwin, Thermo Fisher Scientific). Images of representative areas were acquired on high-resolution plates (Ditabis). For comparison, mouse cremaster tissue containing native blood vessels was imaged as well. The tissue was fixed and processed following a literature protocol by Rehberg et al.[53].

**Statistics and reproducibility**. All analyses were carried out using Microsoft Office Excel (v16.30). Each study was repeated three times independently. The experimental outcomes between independent experiments were in all cases comparable. Statistical significance was analyzed by a two-tailed unpaired Student's *t* test. *P* values < 0.05 were considered statistically significant. All data are presented as mean ±standard deviation.

**Reporting summary**. Further information on research design is available in the Nature Research Reporting Summary linked to this article.

## Data availability

All relevant data supporting the key findings of this study are available within the article and its Supplementary Information files or from the corresponding author upon reasonable request. A reporting summary for this article is available as a Supplementary Information file. Source data are provided with this paper.

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

## Acknowledgements

We thank C. Chen, J. Gautrot, D. Vestweber, R. Adams, I. Bedzhov, G. Trapani, and M. Weiß for constructive discussions and suggestions; C. Chen for providing the molds for the device housing; G. Bixel for providing mouse cremaster tissue; S. Volkery and M. Stasch (BioOptic service unit, MPI Münster), K. Mildner (Electron Microscopy service unit, MPI Münster), S. Winter, and V. Stegemann for providing excellent technical support and C. Brennecka for proofreading the manuscript. This work was financially supported by the Max Planck Society (MPG) (B.T.), the German Research Foundation (DFG) (SFB 1348 A07 (B.T.) and SFB 1009 A02 (L.S.)) and the DFG Cluster of Excellence 'Cells in Motion' (EXC 1003) (L.S. and B.T.). J.L. receives a scholarship from the China Scholarship Council and is supported by the International Max Planck Research School – Molecular Biomedicine, Münster, Germany.

## Author contributions

J.L. and B.T. conceived the study, designed experiments, and interpreted results; J.L. conducted experiments and analyzed data. H.L. synthesized DexVS and established DexVS hydrogel protocol; H.L. characterized hydrogels mechanically; L.S. helped with characterization of in vitro vessels; A.F.B.R. and H.K. provided integrin $\alpha_5\beta_1$-selective ligand; W.J.P. characterized fluid flow profiles; D.Z. performed electron microscopy; J.L. and B.T. wrote the manuscript.

## Funding

## Competing interests

The authors declare no competing interests.
