## [Peer Review File · Nature Communications]

Reviewers' Comments:

Reviewer #1:

Remarks to the Author:

Overview

The manuscript deals with the development of a new in vitro model of angiogenesis. This paper shows how the tunability of synthetic hydrogels could lead to functional new vessels that show the presence of cells with apical-basal polarity, laminin deposition and sufficient degradability (of the material) to allow lumen formation. This manuscript presents data with a new approach that shows significantly improved functional vessel formation compared with previously published studies. However, some of the claims of the authors need further explanation and experimental studies.

Introduction

The introduction states the current challenge faced while developing new biomaterials that promote angiogenesis and the lack of a proper in vitro model for this critical phenomena. However, the advantages of the novel proposed approach cannot be fully appreciated since all the references in the paper but one, are older than 2018. The articles below are some examples of more recent articles that could be used to provide a better state of the art to the reader:

Williams, Nisa P., et al. "Engineering Anisotropic 3D Tubular Tissues with Flexible Thermoresponsive Nanofabricated Substrates." *Biomaterials* 240 (2020): 119856.

Zhang, Guangliang, et al. "ECM Concentration and Cell-Mediated Traction Forces Play a Role in Vascular Network Assembly in 3D Bioprinted Tissue." *Biotechnology and Bioengineering* 117.4 (2020): 1148-1158.

Hu, Michael, et al. "Facile Engineering of Long-Term Culturable Ex Vivo Vascularized Tissues Using Biologically Derived Matrices." *Advanced Healthcare Materials* 7.23 (2018): 1800845.

Results and Discussion

- A rationale for using Dextran MA or Dextran VS for different assays is needed. Currently, the rationale is not strong enough—needs a strong hypothesis.
- The authors claim a different role of the $\alpha v\beta 3$ and $\alpha 5\beta 1$ integrins. However, further experiments such as KO cells, are needed to prove the claim made on page 6: "these experiments suggest that $\alpha v\beta 3$ and not $\alpha 5\beta 1$ integrins are important regulators of multicellular tissue invasion during angiogenic sprouting and that synthetic tissue-engineered constructs must engage $\alpha v\beta 3$ integrins to achieve successful vascularization". Inhibition studies are also needed to show this.
- When comparing the data showed in Figure 4a with the images in Figure 4b, the micrographs are not representative of the difference shown in a.
- Results on the mechanical properties of the hydrogels should be added to the supplementary information.
- Regarding the perfusion through the de novo vessels. Further explanation of the experimental design needs to be provided. Did the authors test if the new vessels resist pressures that are similar to those in vivo? If not, this is a pivotal property to be tested in order to prove functionality and in vivo mimicking.

Material and Methods

- Further explanation of the rationale for the design of the experiments in the choice of 3.5 and 14 days as timepoints is needed.
- Is there any reference in the literature to support that six nuclei is considered multicellular?
- An in-depth explanation of the perfusion experiment needs to be provided.

Reviewer #2:

Remarks to the Author:

This manuscript reports a new synthetic hydrogel system to investigate angiogenesis in vitro. The authors developed an in vitro microfluidic model of angiogenesis that allowed for multicellular invasion in a dextran based hydrogel and ultimately lumen formation with expected apical-basal polarity. Microfluidic device is designed with two parallel channels: coated with HUVECs and flooded with a chemoattractive growth factor cocktail. Dextran hydrogels were either methacrylated (DexMA) or functionalized with vinyl sulfone groups (DexVS) and crosslinked with MMP-sensitive peptides derived from collagen type 1. In this system, they tested the hypothesis that high concentrations of $\alpha 5\beta 1$ integrin engagement are required for early stage angiogenesis. Furthermore, by modulating MMP cleavable sites, they show that gel degradability is crucial for the

development of healthy luminal structures within neo-vessels (grown in synthetic hydrogels respectively). They further confirm group migration is required for neo-vascular formation.

This manuscript is written well overall.

In my opinion, the microfluidic angiogenic model with synthetic gels is excellent. The images are beautiful! However, there were some claims made that demand more evidence, given the data presented (detailed below).

The following concerns need to be addressed in order to strengthen the paper, before it can be considered for publication:

1. Some claims were made in the paper need additional supportive evidence. For example, the authors say that the length and width of their perfusable tubes are inferior to those obtained in natural ECMs, but there are no width measurements presented, and the lengths seem to be limited by the design constraint of the microfluidic device as the vessels were able to extend through the entire length of the devices. The authors are advised to compare widths of lumens since it is included in structural characterization in lumens within synthetic / natural matrices, and clarify the statement that the length of their vessels are different from those in natural ECMs.

2. The authors mentioned that the rate of EC invasion was not different with different concentrations of RGD, but this claim requires additional evidence. For example, in Figure 1d all images were taken at 3.5 days. If the vessels were just invading more slowly, it would be possible that given enough time all RGD concentrations could attain the same degree of vessel formation and multicellularity. To alleviate this concern, measurements should be taken after all sprouts have stopped invading / growing. Contrarily, if sprouts are able to grow / invade until they reach the other side, then cellularity should be counted once all conditions have reached the other side of the microfluidic device. If cellularity is not different at that point, then there is a possibility that different RGD concentration merely leads to different invasion speeds, not necessarily differences in sprouting.

3. Hydrogels were functionalized with varying amounts of RGD to support cell attachment. However, there was no evidence shown for the claim that speed of EC migration was unaffected. Can the authors share this evidence in the Supplemental Information?

4. It is intriguing that the maximum fluorescent bead perfusion was not equivalent to the max width in the microfluidic device. Why won't beads perfuse all the way through the vessels (~780 um)? Are the lumens not open towards the end of the sprouts even after they connect to the other side? This would be a critical capability for this model to function in a clinical tissue engineering application. The authors are advised to address this concern in results and discussion.

5. In the discussion section, the authors note that their results may look contradictory to established theory that $\alpha 5\beta 1$ integrin engagement is required for endothelial cell migration. Although they correctly state that later tube formation allows endothelial cells to deposit endogenous matrix, thus permitting engagement of $\alpha 5\beta 1$, the results shown in Supplementary Figure 3 show redundancy in the roles of both $\alpha 5\beta 1$ and $\alpha v\beta 3$ in cell migration. Further, the images in Figures 1I and 1K look qualitatively comparable and although quantification delineates the two conditions, an analysis before secondary matrix deposition has occurred is needed (and a sprout model with $\alpha v\beta 3$ and $\alpha 5\beta 1$ inhibitors, already validated in the supplementary figures). An earlier time point and inhibitory model would strengthen the authors' point here.

6. Figure 1D: Since all three conditions were evaluated at 3.5 days this alone suggests slower sprouting with less RGD. Does the 0.15 mM RGD concentration group reach the same levels of sprouting as the 6mM group, if allowed to grow for longer periods of time or does growth halt? Quantification of sprout length may help to show that no change in speed of migration occurred. If average sprout length is not different, but the multicellularity is, then it would support their point. However, if length is significantly different, then the authors may need to normalize by sprout length to compare multicellularity. If length is significantly different, then they may also need to consider that migration speed is just different, or show data that proves otherwise.

7. Figure 1I: Why were different concentrations of RGD used compared to initial data? Was the cell distribution in sprouts similar to that seen in 1F? Was the percentage of cells similar to that seen in 1G? Maybe these additional graphs could be included in the supplement if they won't fit in the main figure.

8. Figure 2: Why did the authors use 12 mM CGRGDS and 25.2mM crosslinker concentration? Was that deemed to be the optimal formulation? If so, the authors should show the data that you used to determine optimal formulation.

9. Figure 2D: Are there any zoomed out images of the cell with vacuole showing its place in the sprout?

10. Figure 3A: There seems to be beads where there are no channels. Could the authors explain this issue? In RGD 6 mM there appears to be a sprout that stops and then further down, it looks like the sprout continues with another single cell... The authors are advised to address this issue.

11. Figure 3F: The authors are also advised to quantify max perfusion distance / average perfusion distance like they did earlier in the figure? Then, they could also do a comparison with the regular RGD that was used in Figure 3 A-D, to see if the $\alpha v \beta 3$ selective ligand replicates the results entirely, or if it produces different results in some way.

12. Figure 4B: Are these 3 representative images from each condition or different timepoints? Please add a clarification in the legend.

13. Figure 5A: Why are smaller beads used? Were smaller beads necessary for perfusion throughout the entire length of the channel?

I also have suggestions on some issues related to the presentation and writing of the Introduction section and elsewhere, in the manuscript:

a) The introduction had no indication as to the current landscape of the field. This is not the first time vascular sprouting has been shown in vitro (later mentioned in section 3 of the results page). Gaps in knowledge should be clearly described.

b) There are parts within the introduction, for example, 'synthetic materials are more tunable than natural hydrogels', which are lacking in scientific rigor. The authors further claim that angiogenesis, moreover angiogenic sprouting, is a complex process, yet they fail to expand upon the process in enough detail. For example, the authors did not mention the prevailing theory, $\alpha v \beta 3$ integrin engagement is required for endothelial cell migration, until the results section. Overall, the introduction did not feel appropriately fleshed out.

c) The last sentence of section four of the results should be reworded for clarity and flow.

d) In section two of the results, matrix RGD concentration should be stated and not described as high or low.

e) 'In vivo' should be either italicized or hyphenated.

Great job, overall!

Reviewer #3:

Remarks to the Author:

Major

This manuscript demonstrates the formation of perfusable lumens by culturing HUVECs in a previously-established microfluidic device, but integrated a synthetic hydrogel material. Overall, the experimental design is systematic, and the images support each of the main conclusions.

However, the manuscript lacks novelty in that it does not functionally improve the existing models of in-vitro angiogenic sprouting. The authors don't clearly demonstrate specific advantages or new knowledge gained because of their system, which is largely an integration of a previously developed material and microfluidic assay. Unfortunately, this dampens this reviewers enthusiasm for publication in Nature Communications.

I agree that synthetic hydrogels can provide a more controlled microenvironment, as they do not contain bioactive molecules that can affect cell function in undefinable ways. However, this manuscript does not prove any of these specific benefits. In addition, although the experimerns utilized two types of crosslinkers with different MMP degradability and two types of adhesive peptide ligands, there are numerous publications in the literature that show that these same sequences can improve the collective migration of various cell types. In addition, the experimental design of the microfluidic device and angiogenic sprouting analysis are derivative of work published by the Chen group, except that the authors use of a synthetic hydrogel instead of collagen type 1 (<https://www.pnas.org/content/110/17/6712>, PNAS, 2013). However, the synthetic hydrogel chemistry used was developed by others, so the novelty is simply combining a prior developed material with the methods reported by Chen in 2013.

Minor

- The manuscript emphasizes tailored degradability by using two types of MMP-degradable crosslinkers. However, characterization of the degradability (e.g., kcat and references) is not provided. In addition, the term NCD (natural collagen degradability) is not technically correct. Although the sequence is derived from collagen type 1, it is incorrect to say that it has natural collagen degradability.
- In the manuscript, DexMA hydrogels were used for the multicellularity studies, and DexVS hydrogels were used for lumen formation studies. Because of differences in the chemical reactivity of MA versus VS, this can alter the final network structure and interpretation of some of the results. For overall consistency and interpretation of the results, the authors should use either DexVS or DexMA for all of the experiments. .

A detailed point-by-point response to reviewer concerns follows:

Reviewer #1 (Remarks to the Author):

Overview

The manuscript deals with the development of a new in vitro model of angiogenesis. This paper shows how the tunability of synthetic hydrogels could lead to functional new vessels that show the presence of cells with apical-basal polarity, laminin deposition and sufficient degradability (of the material) to allow lumen formation. This manuscript presents data with a new approach that shows significantly improved functional vessel formation compared with previously published studies. However, some of the claims of the authors need further explanation and experimental studies.

We thank the reviewer for the positive evaluation of our work.

Introduction

*The introduction states the current challenge faced while developing new biomaterials that promote angiogenesis and the lack of a proper in vitro model for this critical phenomena. However, the advantages of the novel proposed approach cannot be fully appreciated since all the references in the paper but one, are older than 2018. The articles below are some examples of more recent articles that could be used to provide a better state of the art to the reader:
Williams, Nisa P., et al. "Engineering Anisotropic 3D Tubular Tissues with Flexible Thermoresponsive Nanofabricated Substrates." *Biomaterials* 240 (2020): 119856.
Zhang, Guangliang, et al. "ECM Concentration and Cell-Mediated Traction Forces Play a Role in Vascular Network Assembly in 3D Bioprinted Tissue." *Biotechnology and Bioengineering* 117.4 (2020): 1148-1158.
Hu, Michael, et al. "Facile Engineering of Long-Term Culturable Ex Vivo Vascularized Tissues Using Biologically Derived Matrices." *Advanced Healthcare Materials* 7.23 (2018): 1800845.*

We thank the reviewer for these thoughtful comments. We have substantially revised the introduction to better highlight the novelty of our work in the context of other published angiogenesis devices. We have also included more recent references to better highlight the current state of the field.

Results and Discussion

1. A rationale for using Dextran MA or Dextran VS for different assays is needed. Currently, the rationale is not strong enough—needs a strong hypothesis.

We apologize for not having provided a rationale for using two different modifications of dextran as hydrogel backbone – this was an oversight and we have now included an explanation in the results section (p. 7). We started our studies using a previously established material system based on methacrylated dextran (DexMA) (Trappmann et al., *Nature Comm* 2017), which allowed us to elucidate the role of matrix adhesiveness on angiogenic sprouting. However, when we extended the culture time to 2 weeks to achieve lumen formation, we noticed that the parent and growth factor source channels started shrinking, and we hypothesized that the gels were softening with time. Nanoindentation measurements confirmed that while hydrogel stiffness remained unchanged during the first days of culture (the time span over which we studied collective endothelial cell migration),

longer incubation periods in cell culture medium would result in lower Young's moduli, most likely due to the hydrolysis of methacrylate groups. Therefore, in order to ensure a constant hydrogel bulk stiffness over the entire culture period, we modified our hydrogels with hydrolytically stable vinyl sulfone groups, that similarly allowed functionalization with adhesive ligands and crosslinking through Michael-type addition. To demonstrate to the reviewer that this approach was successful, we include here the comparison of channel morphology after 1 and 9 days in medium for DexMA and DexVS.

To demonstrate that hydrogel backbone composition itself does not impact multicellular sprouting, we repeated one of the key experiments (impact of RGD concentration on multicellularity of sprouting, initially performed in DexMA hydrogels) in both materials and indeed confirmed that the sprouting phenotype could be replicated in DexVS hydrogels. The images are now included in Supplementary Figure 12.

2. *The authors claim a different role of the $\alpha_v\beta_3$ and $\alpha_5\beta_1$ integrins. However, further experiments such as KO cells, are needed to prove the claim made on page 6: "these experiments suggest that $\alpha_v\beta_3$ and not $\alpha_5\beta_1$ integrins are important regulators of multicellular tissue invasion during angiogenic sprouting and that synthetic tissue-engineered constructs must engage $\alpha_v\beta_3$ integrins to achieve successful vascularization". Inhibition studies are also needed to show this.*

We thank the reviewer for this suggestion and agree that a better characterization of the different roles of $\alpha_v\beta_3$ versus $\alpha_5\beta_1$ integrins is required. While the generation of KO cells is difficult due to low culture passage constraints, we have performed the suggested inhibitor experiments. When HUVECs were induced to sprout into hydrogels functionalized with RGD (a ligand engaging both $\alpha_v\beta_3$ and $\alpha_5\beta_1$ integrins), inhibition of $\alpha_v\beta_3$ integrin engagement through addition of a soluble selective ligand disrupted collective cell migration. However, the inhibition of $\alpha_5\beta_1$ integrin engagement had no effect on the collective migration, and cells still invaded collectively. Together, these experiments confirm the important role of $\alpha_v\beta_3$, but not $\alpha_5\beta_1$ integrins in regulating collective endothelial cell migration during angiogenic sprouting. The data is now shown in Supplementary Figure 10.

3. *When comparing the data showed in Figure 4a with the images in Figure 4b, the micrographs are not representative of the difference shown in a.*

Figure 4a is not a quantification of the images shown in 4b. The purpose of Figure 4a is to introduce the two different degradabilities (NCD and HD) used in Figure 4 by quantifying the resulting difference in cell migration speed. Figure 4b, however, focuses on the characterization of perfusable lumens that have formed after 14 days of culture inside hydrogels of these two different degradabilities. In NCD hydrogels, lumens are short and we therefore only show the sprouts closer to the channel, whereas HD hydrogels support the formation of longer lumen. To demonstrate to the reviewer that in both conditions, HUVECs have indeed migrated all the way to the growth factor source channel, we are including brightfield images of the two conditions below.

4. Results on the mechanical properties of the hydrogels should be added to the supplementary information.

We apologize for this oversight and have now included the mechanical characterization of our hydrogels in Supplementary Figure 18.

5. Regarding the perfusion through the *de novo* vessels. Further explanation of the experimental design needs to be provided. Did the authors test if the new vessels resist pressures that are similar to those *in vivo*? If not, this is a pivotal property to be tested in order to prove functionality and *in vivo* mimicking.

We indeed had not characterized our neovessels with regard to stresses present in blood vessels *in vivo*. To address this concern, we have initiated a collaboration with Prof. William Polacheck (University of North Carolina, now added as a co-author), who calculated shear stresses in the newly formed vessels. The data is now included in Supplementary Fig. 17, and a statement has been added to the manuscript on p. 9. In brief, we introduced flow by hydrostatic pressure and performed time-lapse imaging of fluorescent beads flowing through the neovessels. While the resulting flow velocities ($\sim 100 \mu\text{m/s}$) and wall shear stresses (0.4 Dyn/cm^2) were below the values measured in capillaries *in vivo* (mean blood velocities of $>200 \mu\text{m/s}$ and shear stresses at a minimum of 2.8 Dyn/cm^2 [A. Koutsiaris *et al.*, *Bioreheology* 44, 375-386 (2007)]), this was due to the low driving pressures (17 Pa input pressure). However, we were able to demonstrate that the flow was fully developed as evidenced by the parabolic velocity profile, and are planning to increase pressure to physiologic values in the future. While this requires attaching the device to external active pumps and adapting the ports and device layout to accommodate the fluidics, it is beyond the scope of the current manuscript.

Material and Methods

6. Further explanation of the rationale for the design of the experiments in the choice of 3.5 and 14 days as timepoints is needed.

In order to be able to assess multicellularity, sprouts need to have invaded the gel to a certain minimum extent. We determined that an invasion depth of half way between the parent channel and growth factor source channel was sufficient to reliably analyze multicellularity. Throughout all multicellularity experiments, we therefore fixed cells whenever the leading cells reached the middle position, which was the case after 3.5 days. When we varied hydrogel adhesiveness, sprouting speed was unaffected, so that we fixed all samples after the same culture period. For lumen formation, however, we determined that 2 weeks were needed for the cells to sufficiently remodel the matrix, and hence chose this timepoint. We have now added an explanation regarding timepoints in the methods section (p.17).

7. Is there any reference in the literature to support that six nuclei is considered multicellular?

The definition to consider sprouts containing 6 or more nuclei multicellular is indeed arbitrary. To show that the differences in multicellularity as a function of matrix properties are not only visible if sprouts of 6 or more nuclei are considered multicellular, we have re-analyzed one experiment by considering different numbers of nuclei as multicellular. This analysis shows that the observed trends are independent of the definition of multicellularity, and 6 or more may be taken as a representative. This analysis is now included in Supplementary Figure 4.

8. An in-depth explanation of the perfusion experiment needs to be provided.

We apologize for not articulating the perfusion experiment very clearly, and have now included more details in the methods section of the manuscript (p.19).

To obtain perfusable neovessels, not only lumens must form from the parent channel towards the tip of the sprout, but these lumenized structures must also anastomose with the growth factor channel. This process was particularly difficult to achieve, because relatively thin sprout structures at the forefront had to open up. We therefore extended the culture time from 2 weeks for our standard lumen formation assay to 3 weeks. In addition, since our longest lumenized structures obtained in highly degradable matrices were ca. 500 μm long, we shortened the channel-channel distance to ca. 330 μm . At the end of the experiment, the cells were fixed and the perfusability of the neovessels was tested by aspirating liquid from all reservoirs, and adding a solution of fluorescent beads into the growth factor source channel. Due to differences in hydrostatic pressure between the two channels, beads were able to flow through the neovessels into the parent channel. Importantly, beads did not enter the interstitial hydrogel space, demonstrating that during the process of neovessel formation, the integrity of the host vessel and surrounding matrix stayed intact. We have now added a characterization of the shear stresses present on endothelial cells during this perfusion experiment and compared the values to *in vivo* vessels (see answer to question 5).

Reviewer #2 (Remarks to the Author):

This manuscript reports a new synthetic hydrogel system to investigate angiogenesis in vitro. The authors developed an in vitro microfluidic model of angiogenesis that allowed for multicellular invasion in a dextran based hydrogel and ultimately lumen formation with expected apical-basal polarity. Microfluidic device is designed with two parallel channels: coated with HUVECs and flooded with a chemoattractive growth factor cocktail. Dextran hydrogels were either methacrylated (DexMA) or functionalized with vinyl sulfone groups (DexVS) and crosslinked with MMP-sensitive peptides derived from collagen type 1. In this system, they tested the hypothesis that high concentrations of $\alpha 5\beta 1$ integrin engagement are required for early stage angiogenesis. Furthermore, by modulating MMP cleavable sites, they show that gel degradability is crucial for the development of healthy luminal structures within neo-vessels (grown in synthetic hydrogels respectively). They further confirm group migration is required for neo-vascular formation.

This manuscript is written well overall.

In my opinion, the microfluidic angiogenic model with synthetic gels is excellent. The images are beautiful! However, there were some claims made that demand more evidence, given the data presented (detailed below).

We thank the reviewer for their positive evaluation of our work!

The following concerns need to be addressed in order to strengthen the paper, before it can be considered for publication:

1. Some claims were made in the paper need additional supportive evidence. For example, the authors say that the length and width of their perfusable tubes are inferior to those obtained in natural ECMs, but there are no width measurements presented, and the lengths seem to be limited by the design constraint of the microfluidic device as the vessels were able to extend through the entire length of the devices. The authors are advised to compare widths of lumens since it is included in structural characterization in lumens within synthetic / natural matrices, and clarify the statement that the length of their vessels are different from those in natural ECMs.

To clarify, the statement regarding the length and width of our perfusable tubes being inferior to those in natural ECMs was made in relation to the synthetic gels crosslinked with peptides of native collagen-derived degradability (NCD). Indeed, those vessels were not able to extend throughout the entire length of the device, as has been demonstrated for type I collagen gels (a commonly used model for natural ECMs) (Nguyen et al., PNAS). However, when we increased matrix degradability, vessel formation improved both in terms of length and width. To back up our statement, we have now characterized average and maximum vessel widths in synthetic hydrogels with two different degradabilities, and compared those values to the widths of vessels formed in type I collagen hydrogels. We find that the width of lumens formed in collagen gels can reach up to 50 μm , whereas synthetic hydrogels crosslinked with NCD can only support the formation of up to 20 μm wide tubes. However, increasing matrix degradability leads to vessel widths of up to 55 μm , which is very comparable to what can be achieved in collagen gels. We have now included this characterization in Supplementary Fig. 16 and use the data to clarify the statement above (p. 8/9).

2. The authors mentioned that the rate of EC invasion was not different with different concentrations of RGD, but this claim requires additional evidence. For example, in Figure 1d all images were taken at 3.5 days. If the vessels were just invading more slowly, it would be possible that given enough time all RGD concentrations could attain the same degree of vessel formation and multicellularity. To alleviate this concern, measurements should be taken after all sprouts have stopped invading / growing. Contrarily, if sprouts are able to grow / invade until they reach the other side, then cellularity should be counted once all conditions have reached the other side of the microfluidic device. If cellularity is not different at that point, then there is a possibility that different RGD concentration merely leads to different invasion speeds, not necessarily differences in sprouting.

The initial cell migration speed (at 3.5 days, the point at which cells have reached the middle position between the two channels and at which we evaluate multicellularity) did not differ between matrices with different concentrations of tethered RGD. While there was intrinsic variation from experiment to experiment, we have analyzed the migration speed for 3 independent repeats and have now included the data in Supplementary Figure 2. Based on this data, we can conclude that the changes in multicellularity with matrix adhesiveness are not a consequence of different cell migration speeds.

Nonetheless, the reviewer's concern that differences in multicellularity could potentially only occur during early sprouting, is valid. To address this point, we have extended the culture time until the cells reached the other channel, and analyzed multicellularity again. We still find the same trend, with cells migrating collectively at high RGD concentrations only. At lower RGD concentrations, cells eventually stopped invading, possibly as a result of cell death due to lack of integrin activation. We have now included this data in Supplementary Figure 3 to show conclusively that the differences in migration mode due to matrix adhesiveness are independent of the sprouting stage.

3. Hydrogels were functionalized with varying amounts of RGD to support cell attachment. However, there was no evidence shown for the claim that speed of EC migration was unaffected. Can the authors share this evidence in the Supplemental Information?

A quantification of cell migration speed through matrices functionalized with different concentrations of RGD has now been included as Supplementary Figure 2 (see explanation above).

4. It is intriguing that the maximum fluorescent bead perfusion was not equivalent to the max width in the microfluidic device. Why won't beads perfuse all the way through the vessels (~780 um)? Are the lumens not open towards the end of the sprouts even after they connect to the other side? This would be a critical capability for this model to function in a clinical tissue engineering application. The authors are advised to address this concern in results and discussion.

For all experiments in which we studied the impact of matrix adhesiveness and degradability on lumen formation, we stopped the cultures after 2 weeks, at which point the lumens had not opened throughout the entire length of the sprouts yet. The process of lumen opening turned out to be relatively slow, in particular towards the tip of the sprouts where the structures were thinner (possibly due to the absence of a growth factor gradient closer to

the source channel). Therefore, in order to prove that anastomosis with the source channel is possible, we extended the culture time and also shortened the channel-channel distance to 330 μm , and could indeed achieve fully perfused vessels. As a side note: the fully perfusable distance that has been achieved in type I collagen gels with the same device (Nguyen, PNAS 2013), is in a similar same range (ca. 380 μm). In collagen gels, HUVECs digested the matrix, thereby shortening the distance between the two channels.

5. In the discussion section, the authors note that their results may look contradictory to established theory that $\alpha_5\beta_1$ integrin engagement is required for endothelial cell migration. Although they correctly state that later tube formation allows endothelial cells to deposit endogenous matrix, thus permitting engagement of $\alpha_5\beta_1$, the results shown in Supplementary Figure 3 show redundancy in the roles of both $\alpha_5\beta_1$ and $\alpha_v\beta_3$ in cell migration. Further, the images in Figures 1I and 1K look qualitatively comparable and although quantification delineates the two conditions, an analysis before secondary matrix deposition has occurred is needed (and a sprout model with $\alpha_v\beta_3$ and $\alpha_5\beta_1$ inhibitors, already validated in the supplementary figures). An earlier time point and inhibitory model would strengthen the authors' point here.

We would like to clarify that Supplementary Figure 6 (previously Supplementary Figure 3) shows redundancy in the roles of integrins $\alpha_v\beta_3$ and $\alpha_5\beta_1$ for **cell adhesion and spreading**, not migration. In contrast, the angiogenic sprouting assays in Figures 1i and 1k demonstrate that the matrix tethering of $\alpha_5\beta_1$ selective ligands does not support collective migration, whereas binding of $\alpha_v\beta_3$ selective ligand promotes collective cell migration in a concentration dependent manner. We agree that the differences in multicellularity may have been difficult to identify from the images provided and have now optimized the brightness and contrast of the fluorescent images.

In addition, we have followed the excellent suggestion by the reviewer to perform integrin inhibition experiments that further confirm the distinct roles of $\alpha_v\beta_3$ and $\alpha_5\beta_1$ integrins. In particular, we allowed HUVECs to invade into a hydrogel functionalized with RGD (a ligand engaging both $\alpha_v\beta_3$ and $\alpha_5\beta_1$ integrins) and inhibited $\alpha_v\beta_3$ and $\alpha_5\beta_1$ integrin engagement, respectively, by using specific ligands. While the inhibition of $\alpha_5\beta_1$ integrins had no effect on collective HUVEC migration, inhibition of $\alpha_v\beta_3$ integrins disrupted multicellularity. Together, these experiments clearly demonstrate that $\alpha_v\beta_3$, but not $\alpha_5\beta_1$ integrins regulate endothelial sprout multicellularity. The data is now shown in Supplementary Figure 10.

6. Figure 1D: Since all three conditions were evaluated at 3.5 days this alone suggests slower sprouting with less RGD. Does the 0.15 mM RGD concentration group reach the same levels of sprouting as the 6mM group, if allowed to grow for longer periods of time or does growth halt? Quantification of sprout length may help to show that no change in speed of migration occurred. If average sprout length is not different, but the multicellularity is, then it would support their point. However, if length is significantly different, then the authors may need to normalize by sprout length to compare multicellularity. If length is significantly different, then they may also need to consider that migration speed is just different, or show data that proves otherwise.

The speed of HUVEC migration is independent of the concentration of matrix bound RGD. While there is intrinsic variation between individual experiments, there was no statistical difference in speed (see answer to question number 2). We realized that the picture chosen

for Figure 1d was confusing, as the cells sprouting through hydrogels functionalized with 0.15 mM RGD indeed had not invaded as far as in the 6 mM condition. We have now exchanged this picture to better represent our statistical measurements that yielded equal migration speed across all RGD concentrations used. The data regarding migration speed is now included in Supplementary Figure 2.

7. Figure 1I: Why were different concentrations of RGD used compared to initial data? Was the cell distribution in sprouts similar to that seen in 1F? Was the percentage of cells similar to that seen in 1G? Maybe these additional graphs could be included in the supplement if they won't fit in the main figure.

In the initial data (Figure 1d), linear RGD was used as an adhesive ligand that activates several integrins, in particular $\alpha_v\beta_3$ and $\alpha_5\beta_1$ integrins. However, in Figures 1i and 1k, the effects of varying concentrations of $\alpha_v\beta_3$ and $\alpha_5\beta_1$ integrin specific ligands on the multicellularity of angiogenic sprouting were tested. Since all three ligands have different binding affinities to the respective integrins, we first tested at which concentrations the ligands were able to activate integrins sufficiently to support cell spreading (Supplementary Figure 7a, b). These concentrations were then taken as highest conditions for the multicellularity assays. We apologize for not having clearly articulated our approach in the previous version of the manuscript, which has now been improved (see p.16 of Materials and Methods).

We have now included the analysis of 'cell distribution in sprouts' and '% cells in multicellular sprouts' for the conditions in Figures 1i and 1k as Supplementary Figure 9.

8. Figure 2: Why did the authors use 12 mM CGRGDS and 25.2mM crosslinker concentration? Was that deemed to be the optimal formulation? If so, the authors should show the data that you used to determine optimal formulation.

Based on the data in Figure 1d, RGD concentrations above 6 mM were able to promote multicellular sprouting. Concentrations higher than 6 mM would neither improve, nor disrupt collectivity. In order to ensure the presence of sufficient amounts of adhesive ligands even after longer culture times, we chose 12 mM for all long-term lumen experiments. The choice of a crosslinker concentration of 25.2 mM was based on our recent paper (Trappmann et al., Nature Comm 2017), in which we studied the role of matrix stiffness on the collective migration of endothelial cells during angiogenic sprouting. In this study, concentrations of 25 mM crosslinker were optimal to induce collective migration.

9. Figure 2D: Are there any zoomed out images of the cell with vacuole showing its place in the sprout?

The zoomed out image of the cell (indicated by an arrow) with the vacuole shown in Figure 2 is pasted here (scale bar 100 μm):

10. Figure 3A: There seems to be beads where there are no channels. Could the authors explain this issue? In RGD 6 mM there appears to be a sprout that stops and then further down, it looks like the sprout continues with another single cell... The authors are advised to address this issue.

It is indeed correct that in the condition with low concentrations of matrix bound RGD (0.15 mM) beads can enter the gel in areas without endothelial cell tubes. This is due to the setup of the hydrogel system, in which endothelial cells permanently cleave the matrix, thereby leaving behind tracks that beads can enter. To alleviate the concern that in the high adhesiveness, multicellular conditions beads only enter through cell cleaved tracks, but not actual lumen, we had included vertical and horizontal sections of the 3D sprouts showing the actual location of beads inside the lumen (Figure 2 bi and bii).

Regarding the picture in Figure 3A, 6 mM RGD condition: we agree that in these maximum projection images, the orientation of the sprouts relative to each other is difficult to visualize. We have therefore checked the specific image in 3D, and realized that the sprout and single cell that the reviewer pointed out are in fact two separate sprouts that just coincidentally follow the same direction.

11. Figure 3F: The authors are also advised to quantify max perfusion distance / average perfusion distance like they did earlier in the figure? Then, they could also do a comparison with the regular RGD that was used in Figure 3 A-D, to see if the $\alpha v\beta 3$ selective ligand replicates the results entirely, or if it produces different results in some way.

We have added some additional characterizations of the lumens formed within matrices of different types of adhesive ligands (Supplementary Figure 14). The comparison of lumen

metrics between hydrogels functionalized with 12 mM RGD and 1.5 mM of the $\alpha_v\beta_3$ integrin selective ligand resulted in no statistical differences (Supplementary Figure 15), suggesting that the $\alpha_v\beta_3$ selective ligand indeed replicates the RGD results entirely. We believe that this is a very important statement, which has now been added to the manuscript (p.8) and thank the reviewer for this suggestion.

12. *Figure 4B: Are these 3 representative images from each condition or different timepoints? Please add a clarification in the legend.*

The images shown in Figure 4b are indeed 3 representative images each for lumens in NCD and HD hydrogels, all taken after the same culture period of 14 days. This has now been clarified in the figure legend.

13. *Figure 5A: Why are smaller beads used? Were smaller beads necessary for perfusion throughout the entire length of the channel?*

We indeed used smaller (1 μm) beads to ensure full perfusion throughout the entire length of the channel. However, 4 μm beads were able to flow through the channels as well, but would occasionally result in the blockage of flow, and were therefore not ideal to visualize fluid flow profiles (Supplementary Figure 17).

I also have suggestions on some issues related to the presentation and writing of the Introduction section and elsewhere, in the manuscript:

a) The introduction had no indication as to the current landscape of the field. This is not the first time vascular sprouting has been shown in vitro (later mentioned in section 3 of the results page). Gaps in knowledge should be clearly described.

We agree that the introduction did not clearly describe the current landscape of the field. We have therefore substantially rewritten the introduction to clearly state the gaps in knowledge and how we intend to fill them with our study.

b) There are parts within the introduction, for example, 'synthetic materials are more tunable than natural hydrogels', which are lacking in scientific rigor. The authors further claim that angiogenesis, moreover angiogenic sprouting, is a complex process, yet they fail to expand upon the process in enough detail. For example, the authors did not mention the prevailing theory, $\alpha_v\beta_3$ integrin engagement is required for endothelial cell migration, until the results section. Overall, the introduction did not feel appropriately fleshed out.

We thank the reviewer for this important criticism and have expanded the introduction to better introduce the role of matrix adhesiveness and $\alpha_v\beta_3$ integrin engagement in angiogenesis, which is indeed central to our study.

c) The last sentence of section four of the results should be reworded for clarity and flow.

We fully agree with the reviewer that the mentioning of intracellular vacuoles at the end of the paragraph disrupted the flow of the story. We have therefore removed the sentence and incorporated this aspect into an earlier sentence (p.7).

d) *In section two of the results, matrix RGD concentration should be stated and not described as high or low.*

We apologize that we did not state the actual matrix bound RGD concentrations used, which has now been corrected (p.5).

e) *'In vivo' should be either italicized or hyphenated.*

This has now been corrected throughout the manuscript.

Great job, overall!

Reviewer #3 (Remarks to the Author):

Major

This manuscript demonstrates the formation of perfusable lumens by culturing HUVECs in a previously-established microfluidic device, but integrated a synthetic hydrogel material. Overall, the experimental design is systematic, and the images support each of the main conclusions. However, the manuscript lacks novelty in that it does not functionally improve the existing models of in-vitro angiogenic sprouting. The authors don't clearly demonstrate specific advantages or new knowledge gained because of their system, which is largely an integration of a previously developed material and microfluidic assay. Unfortunately, this dampens this reviewers enthusiasm for publication in Nature Communications.

We thank the reviewer for bringing up this important criticism and apologize for not having clearly articulated the main aim of our study. The microfluidic device, as well as the synthetic hydrogel system have been published by the Chen lab before, and it was not our intention to claim novelty in these tools. Instead, the novelty of our study lies in the generation of a fully functional, i.e. perfusable vasculature that is connected to a host vessel in a synthetic material *in vitro*, which has not been achieved before. One of the major challenges that the tissue engineering field faces is the lack of materials that support angiogenesis, because the design criteria needed for their fabrication are unknown. This knowledge gap is mainly caused by a lack of understanding of how individual matrix properties affect angiogenesis. To fill this gap, we have established a model system that is based on a synthetic hydrogel matrix with tunable biochemical and mechanical properties, while at the same time mimicking the structural features of *in vivo* angiogenesis. Using this model, we uncover the roles of matrix adhesiveness and degradability in guiding distinct stages (multicellular invasion versus lumen formation) of angiogenic sprouting. By tuning these two matrix properties, we were for the first time able to achieve the formation of fully perfused vascular lumens in a synthetic hydrogel *in vitro*. Most importantly, we believe that this model will enable the screening of other ECM parameters and their roles in angiogenesis in the future, thereby constituting an important step towards design criteria for the generation of synthetic tissue engineered constructs that allow for vascularization.

We have now rewritten the introduction substantially to better capture the current state of the art in the field, clearly articulate the knowledge gap and state how our first *in vitro*

model of lumen formation in synthetic hydrogels can contribute to the future design of vascularized tissues.

I agree that synthetic hydrogels can provide a more controlled microenvironment, as they do not contain bioactive molecules that can affect cell function in undefinable ways. However, this manuscript does not prove any of these specific benefits.

The choice of a synthetic hydrogel with controllable microenvironmental properties is essential to the development of a functional model to screen how individual ECM properties regulate angiogenesis. We believe that the lack of such model systems poses a great obstacle towards the design of materials for vascularization, as currently, it is impossible to test how individual matrix properties regulate angiogenesis in settings that capture all the structural and anatomical features of *in vivo* vessels. In fact, we have used the tunability of the incorporated hydrogel model to screen the impact of matrix adhesiveness on angiogenic sprouting, and show that only if matrices are presented with adhesive ligands activating $\alpha_v\beta_3$ integrins, multicellular endothelial cell sprouting can be obtained in a concentration-dependent manner, which in turn enables the later formation of lumens. This observation does not only provide a design parameter for vascularizable materials, but it also gives insights into the previously unrecognized role of $\alpha_v\beta_3$ integrins in regulating the collectivity of angiogenic sprouting. Additionally, we have tested the effect of matrix degradability by exchanging the crosslinker sequence in our tunable hydrogel system, and found that only if matrix degradability was sufficiently high, lumens with sizes similar to those achieved in hydrogels based on natural ECM proteins could be achieved. Together, we have demonstrated the benefits of incorporating a tunable hydrogel system throughout the manuscript; in fact, the novel mechanistic insights gained from our studies were only enabled by the tunable nature of the hydrogel.

In addition, although the experimernts utilized two types of crosslinkers with different MMP degradability and two types of adhesive peptide ligands, there are numerous publications in the literature that show that these same sequences can improve the collective migration of various cell types. In addition, the experimental design of the microfluidic device and angiogenic sprouting analysis are derivative of work published by the Chen group, except that the authors use of a synthetic hydrogel instead of collagen type 1 (<https://www.pnas.org/content/110/17/6712>, PNAS, 2013). However, the synthetic hydrogel chemistry used was developed by others, so the novelty is simply combining a prior developed material with the methods reported by Chen in 2013.

Here, we developed an *in vitro* model of fully functional vasculature in a synthetic hydrogel with tunable biochemical and mechanical properties. While this model indeed incorporates some tools that have been published before (and we did not intend to claim novelty for these), it is the first example of perfusable neovessels that are connected to the host vasculature inside synthetic materials *in vitro*. Using this model, we demonstrate for the first time how individual matrix properties regulate collective cell migration and lumen formation during angiogenesis (that had previously not been studied due to the lack of a model system), thereby highlighting the importance of such a model system for the rational design of tissue engineered constructs in future applications.

Minor

1. The manuscript emphasizes tailored degradability by using two types of MMP-degradable crosslinkers. However, characterization of the degradability (e.g., kcat and references) is not provided. In addition, the term NCD (natural collagen degradability) is not technically correct. Although the sequence is derived from collagen type 1, it is incorrect to say that it has natural collagen degradability.

The crosslinker sequences have been commonly used in the literature and are very well characterized. The kcat values and references have been included in Supplementary Table 2.

We agree with the reviewer that the term ‘native collagen degradability’ is not fully correct, and we have therefore exchanged it with ‘native collagen-derived degradability’ throughout the manuscript.

2. In the manuscript, DexMA hydrogels were used for the multicellularity studies, and DexVS hydrogels were used for lumen formation studies. Because of differences in the chemical reactivity of MA versus VS, this can alter the final network structure and interpretation of some of the results. For overall consistency and interpretation of the results, the authors should use either DexVS or DexMA for all of the experiments.

We fully agree with the reviewer that using two different hydrogel backbone materials could complicate the interpretation of the results, and we apologize for not making our rationale for using these two modifications clear: We started our studies using a previously established material system based on methacrylated dextran (DexMA) (Trappmann et al., Nature Comm 2017), which allowed us to elucidate the role of matrix adhesiveness on angiogenic sprouting. However, when we extended the culture time to 2 weeks to achieve lumen formation, we noticed that the hydrogel channels started shrinking, and we hypothesized that the gels were softening with time. Nanoindentation measurements confirmed that while hydrogel stiffness remained unchanged during the first days of culture (the time span over which we studied collective endothelial cell migration), longer incubation periods in cell culture medium would result in lower Young’s moduli, most likely due to the hydrolysis of methacrylate groups. Therefore, in order to ensure a constant hydrogel bulk stiffness over the entire culture period, we modified our hydrogels with hydrolytically stable vinyl sulfone groups, that similarly allowed functionalization with adhesive ligands and crosslinking through Michael-type addition. To alleviate the reviewer’s concern that hydrogel backbone composition itself could impact multicellular sprouting, we repeated one of the key experiments (impact of RGD concentration on multicellularity of sprouting, initially performed in DexMA hydrogels) in both materials and indeed confirmed that the sprouting phenotype could be replicated in DexVS hydrogels. The images are now included in Supplementary Figure 12.

Reviewers' Comments:

Reviewer #1:

Remarks to the Author:

Although the authors have attempted to respond to the queries-there are some critical studies that need to be conducted.

1. Specifically, KO studies of the integrin sequences are needed and merely inhibitor studies are not sufficient due to the lack of specificity.
2. Perfusable constructs are only an increment in the current state of art. The cell type used is HUVECs--which is a far irrelevant cell type used. Microvascular endothelial cells (+KO) should have been studied in addition at least for validation. The conclusions of this study cannot be stated the way they are framed due to the cell type studied.

Reviewer #2:

Remarks to the Author:

Thank you for the revised manuscript and for addressing the feedback on the previous version!

Great job, overall! And good luck with future projects!

A detailed point-by-point response to reviewer concerns follows:

Reviewer #1 (Remarks to the Author):

Although the authors have attempted to respond to the queries-there are some critical studies that need to be conducted.

1. Specifically, KO studies of the integrin sequences are needed and merely inhibitor studies are not sufficient due to the lack of specificity.

We agree with the reviewer that KO studies of specific integrins would further support our finding that $\alpha_v\beta_3$, and not $\alpha_5\beta_1$ integrins, regulate multicellular migration during angiogenic sprouting. However, due to technical difficulties, this experiment is not feasible: seeding of endothelial cells inside the parent vessel requires strong integrin activation, as cells have to attach to the hydrogel matrix while flowing through the channel. Due to the impaired adhesivity of integrin KO cells, cell attachment is delayed and channel seeding impossible. Therefore, we addressed the reviewer's concern by using selective integrin inhibitors, which were added after channels were endothelialized. The chosen inhibitors against $\alpha_v\beta_3$ and $\alpha_5\beta_1$ integrins (developed in the laboratory of Prof. Horst Kessler) have been characterized extensively and most importantly, proved to be specific against the two integrins (Kapp, T.G. *et al.* A Comprehensive Evaluation of the Activity and Selectivity Profile of Ligands for RGD-binding Integrins. *Sci Rep* 7, 39805 (2017)). Using these inhibitors, we could clearly demonstrate that $\alpha_v\beta_3$ integrin activation regulates the multicellularity of angiogenic sprouts, whereas $\alpha_5\beta_1$ does not. This data complements our initial studies, in which cells were only able to sprout collectively, if hydrogels were functionalized with immobilized $\alpha_v\beta_3$, but not $\alpha_5\beta_1$ selective ligands.

Together, by perturbing integrin-matrix interactions through 1) altering matrix adhesiveness and 2) integrin signaling, we could clearly uncover the important role of $\alpha_v\beta_3$ integrins in regulating multicellular migration during angiogenic sprouting.

2. Perfusable constructs are only an increment in the current state of art. The cell type used is HUVECs--which is a far irrelevant cell type used. Microvascular endothelial cells (+KO) should have been studied in addition at least for validation. The conclusions of this study cannot be stated the way they are framed due to the cell type studied.

We agree with the reviewer that microvascular endothelial cells are a more relevant cell type with respect to tissue engineering applications. Following this reviewer's concern, we have therefore replicated our two key experiments with microvascular endothelial cells:

1) We tested how human lung microvascular endothelial cells (HMVECs) respond to different concentrations of matrix adhesive ligands. Indeed, if HMVECs were induced to sprout through matrices functionalized with high concentrations of the adhesive ligand RGD, multicellular structures were obtained, in contrast to matrices with low concentrations of coupled RGD, which only supported single cell migration. This result clearly demonstrates that HMVECs respond to changes in matrix adhesiveness in a similar manner to HUVECs. The data is now included as Supplementary Figure 6 and a statement has been added to the results section of the manuscript (p.6).

2) Additionally, we have attempted to replicate lumen formation within hydrogels of optimized adhesiveness and degradability using microvascular endothelial cells. While we were able to achieve the formation of perfusable lumen (with the correct apical-basal

polarity and deposition of the basement membrane protein laminin) also with HMVECs, the width and length of the structures were inferior to those obtained with HUVECs.

The obstacle that HMVECs yield less perfusable vessel networks seems to be commonly faced in the literature, even when most optimal hydrogels based on natural ECM proteins are employed. As an example, we are citing from a recent Nature Protocols paper by the laboratory of Prof. Roger Kamm, one of the pioneers in the development of microfluidic devices to model angiogenesis *in vitro*: ‘While it is possible for human microvascular endothelial cells (HMVECs) to be used instead of HUVECs, we have observed that the current protocol with HMVECs yields significantly less perfusable and less interconnected vascular networks. Further optimization of the initial ECM composition and exogenous growth factors is required if HMVEC culture is desired.’ (cited from Chen, M.B. *et al.*, *Nature Protocols* 12, 865-880 (2017))

Based on our experience, one problem could be the lack of suitable commercial culture media that best support the growth of HMVECs. In particular, while setting up our assay, we screened several types of media from different suppliers and noticed that none supported optimal proliferation levels of HMVECs, which are required for the formation of long and wide lumens (and the level of cell proliferation varied between media from different suppliers).

Despite these shortcomings, we were able to achieve lumen formation and most importantly, could demonstrate that the determined matrix design criteria are not exclusive to the usage of HUVECs, but extend to microvascular endothelial cells resident in *in vivo* tissues as well.

Reviewer #2 (Remarks to the Author):

Thank you for the revised manuscript and for addressing the feedback on the previous version!

Great job, overall! And good luck with future projects!

We thank the reviewer for the very positive evaluation of our work and in particular, for their helpful and most constructive comments that have greatly enhanced the quality of this manuscript.

Reviewers' Comments:

Reviewer #1:

Remarks to the Author:

The new inhibitor data is not clear in the revised manuscript. The results and discussion is missing. In addition, the subsequent claims of not using KO needs to be adjusted.

I am satisfied with the second concern of the validation with a relevant cell type.

A detailed point-by-point response to reviewer concerns follows:

Reviewer #1 (Remarks to the Author):

The new inhibitor data is not clear in the revised manuscript. The results and discussion is missing. In addition, the subsequent claims of not using KO needs to be adjusted.

We would like to point out that the inhibitor studies mentioned in our last ‘point-by-point response’ had been performed during the first round of revisions, and are included as Supplementary Figure 11 and on page 6 of the main text. We apologize for not having communicated this clearly in our last response.

Regarding the second point, we are not sure which claims regarding KO cells the reviewer is asking us to adjust, since we did not include any such claims in the manuscript. In our last ‘point-by-point response’, we explained that the use of integrin KO cells was not possible in our devices due to technical limitations, and we therefore reverted to inhibitor studies, which we then included in our manuscript.

I am satisfied with the second concern of the validation with a relevant cell type.

We are glad that the reviewer is satisfied with our validation experiments. We would like to thank them for the thoughtful assessment of our manuscript and the constructive comments which have significantly improved the quality of our study.